# Breaking down barriers on PV trade will facilitate global carbon mitigation

Mudan Wang [1,2], Xianqiang Mao [1,2✉], Youkai Xing[1,2,8], Jianhong Lu[1,2], Peng Song [3✉], Zhengyan Liu[2,4], Zhi Guo[1,2], Kevin Tu[5] & Eric Zusman[6,7]

The global trade of solar photovoltaic (PV) products substantially contributes to increases in solar power generation and carbon emissions reductions. This paper depicts global PV product trade patterns, explores emissions reduction potential, and evaluates the impeding effect of tariff barriers on global PV product trade and emissions reductions. Solar power generation will result in a reduction of emissions in a range of 50–180 gigatons of carbon dioxide equivalent ($GtCO_2e$) between 2017 and 2060 in a business as usual (BAU) scenario. Compared with BAU, during 2017–2060, global total solar cell and module production and installation will increase by roughly 750 gigawatts (GW) if half of the status quo trade barrier are removed, while it will decrease by 160–370 GW under tensioned trade barrier scenarios. Trade barrier reduction by half from the 2017 status quo level will increase the net carbon emissions mitigation potential by 4–12 $GtCO_2e$ by 2060, while extra trade barrier imposition will result in global net carbon emissions mitigation potential decreasing by up to 3–4 $GtCO_2e$ by 2060. Well-coordinated policy and institutional reforms are recommended to facilitate PV product trade and to deliver the related global environmental benefits.

[1] School of Environment, Beijing Normal University, Xinjiekouwai Street No. 19, Beijing 100875, PR China. [2] Center for Global Environmental Policy, Beijing Normal University, Xinjiekouwai Street No. 19, Beijing 100875, PR China. [3] School of Public Policy and Administration, Chongqing University, Shazheng Street No. 174, Chongqing 400044, PR China. [4] Institute of Spatial Planning and Regional Economy, China Academy of Macroeconomic Research, Beijing 100038, P. R. China. [5] Center on Global Energy Policy at Columbia University SIPA, 1255 Amsterdam Avenue, New York, NY 10027, USA. [6] Institute for Global Environmental Strategies, 2108-11 Kamiyamaguchi Hayama, Kanagawa, Japan. [7] National Institute for Environmental Studies, 16-2 Onogawa, Tsukuba, Ibaraki, Japan. [8] Present address: Transport Planning and Research Institute, Ministry of Transport, No.6, Shuguangxilijia, Chaoyang District, Beijing 100028, China. ✉email: maoxq@bnu.edu.cn; songpeng_ee@cqu.edu.cn

The importance of reducing greenhouse gases (GHGs) to net zero to limit global warming within 1.5 °C has been well recognized, and the Intergovernmental Panel on Climate Change (IPCC) predicts that 38–88% of primary energy and 59–97% of electricity must come from renewable resources by 2040–2055 to achieve that goal[1]. Solar power is expected to play a key role in facilitating low-carbon transitions, mitigating climate change and meeting energy demands[2–4]. During 2007–2017, the global annual installed solar photovoltaic (PV) power capacity increased from 2 gigawatts (GW) to 103 GW, and global PV cumulative installations increased from 8 GW to 409 GW[5]. In 2007, global PV power generation was 7.48 terawatt hours (TWh), only 0.04% of total production, but it surged to 443.55 TWh in 2017, accounting for 1.72% of the total amount[6]. Rapid development of international trade contributes substantially to global PV product production and application expansion by providing less expensive PV products and reducing costs. The traded capacities of solar cells and modules have reached 79.65 GW in 2017, accounting for 19.47% of the global cumulative PV capacity installation in that year. Almost 76.89% of the newly installed global capacity in 2017 is related to traded solar cells and modules, and this proportion is 96.19% in 2018[7].

Although PV power generation is nearly 'zero emissions' during operation and could indeed help to substantially reduce carbon emissions[8–13], its emissions should not be ignored when the whole life cycle of PV products is considered[14–18]. Studies have been conducted ranging from calculating the life cycle emissions and emissions reduction potential of PV products[8,10–16,19–24] to PV systems including balance of system (BOS) and storage and/or batteries[25–29] and their end-of-life emission contribution[30,31], life cycle impact comparisons between solar power and other power generation technologies[32,33], and the impacts of PV application on future electricity grids[34,35] in various geographical areas.

Previous studies have also addressed the rapid development of international trade and production specialization[36,37], PV product trade patterns and structures[38], and emissions embodied in and the environmental impacts of the PV product trade[9,23,24,39]. With international solar PV product market expansion, protectionism has grown in the form of antidumping or countervailing measures regarding PV modules. Recent examples of these tensions include conflicts between the USA and China, the EU and China, etc.[5,40] In 2017, the USA initiated the "301 Investigation" against China on imported goods (solar panels included) and led to a series of conflicting trade measures between the two globally largest trade partners, spilling over to other economies, including the EU, Canada and Mexico[41–45]. Most studies have concentrated on the economic impact of the USA–China trade war[41,43,46–52], and only a few have noted that trade wars are also likely to affect the environment by changing the global supply and consumption chain[53,54] and are less conducive to clean energy development in less-developed regions[55]. Very few studies have paid attention to the negative effects of trade wars on PV product trade, production and application[56], cutting into global emissions reduction potential.

In this work, we aim to explore the impacts of trade liberalization and restriction measures on PV products, which could affect global PV trade, production, installation, clean power generation and carbon emissions reduction potential. This study constructs a trade flow matrix (TFM) of PV products first. Then, embodied carbon in the PV product trade are calculated, and the net emissions reduction potential of globally traded PV product applications is projected. Finally, this study attempts to disclose how trade barriers could cause global carbon mitigation potential to deteriorate. Here we show that, in 2017, although roughly 0.13 gigatons of carbon dioxide equivalent (GtCO₂e) is embodied in the global PV product trade, the application of traded solar cells and modules results in 30-year lifetime net emissions reductions of up to 1.6 GtCO₂e. Solar power generation will result in a reduction of emissions in a range of 50–180 GtCO₂e between 2017 and 2060 in BAU scenario. Simulation results clearly show that trade protectionism harms not only the global PV product trade but also the environment. Removal of half of status quo trade barriers will increase PV applications by 7.2% and improve cumulative net carbon emissions reduction potential by 4–12 GtCO₂e, while extra trade barrier impositions will result in global PV applications decreasing by 1.6% to 3.5% and cumulative net carbon emissions mitigation potential decreasing by up to 3–4 GtCO₂e by 2060.

## Results

**Status quo global PV product trade patterns and embodied carbon flows.** The global PV product trade pattern are presented by TFMs. In this study, TFMs based on export and import trade values are constructed for 2017 (see the 'Methods' section and Supplementary Information 2 for details), which indicates a significant role for international trade in promoting global PV power application (see Data sheet 1 in Source data for more details). This study mainly adopts export values to construct TFMs and to perform trade flow calculations and analyses (shown in Fig. 1) for various regions and countries/economies (see Table 1 for detailed region descriptions).

East Asia ranks as the top PV product exporter region at US $23.94 billion, accounting for 61.92% of the global total, followed by Europe (US$7.07 billion) and Southeast Asia (US$5.00 billion). Specifically, China is the largest exporting country, with US$14.04 billion, accounting for 36.32% of the global total, followed by Japan (US$3.69 billion). For PV product trade inflows, East Asia is also the largest importer region, with US$13.29 billion or ~34.39% of the global total, followed by Southeast Asia (US$7.64 billion, 19.76%) and Europe (US$6.81 billion, 17.62%). China, the USA and Japan are the top 3 importing countries, accounting for 31.60% of the global total aggregately. Intra-East Asia trade stands out at US$8.25 billion. East Asia-Europe trade and East Asia-Americas trade are also important, amounting to US$1.65 billion and US$3.14 billion, respectively. At the country/economy level, the largest bilateral export flow is from China to India, at US$2.72 billion (see Data sheet 2 in Source data for more details)

Regarding specific PV product categories, in 2017, East Asia, America and Europe are major silicon supplier regions, accounting for more than 95% of the global share. The USA, Republic of Korea and Germany each account for more than 20% of the global export market share. East Asia, Europe, and Southeast Asia are the top 3 exporter regions of silicon wafers, with a total global export share of more than 91%. China and Japan are the most important silicon wafer providers, accounting for more than 56% of the total silicon wafer exports. East Asia, Europe, and Southeast Asia are also the top 3 exporters of solar cells and modules, accounting for more than 98% of the total exports. China, contributing more than 45% of the trade value, is the largest exporting country of solar cells and modules. For PV products inflows, East Asia is the largest importer region of silicon (83.01% of the global total) and silicon wafers (52.19% of the global total). Southeast Asia also stands out in silicon and silicon wafer imports, accounting for 8.94% and 22.22%, respectively. For solar cells and modules, Europe, Southeast Asia, and the Americas are the main importer and consumer regions, constituting 22.76%, 20.94% and 18.72% of the global total, respectively. The USA, India and Japan are the top 3 solar cells and modules importing countries, the imports of which aggregately account for 33.60% of the global total.

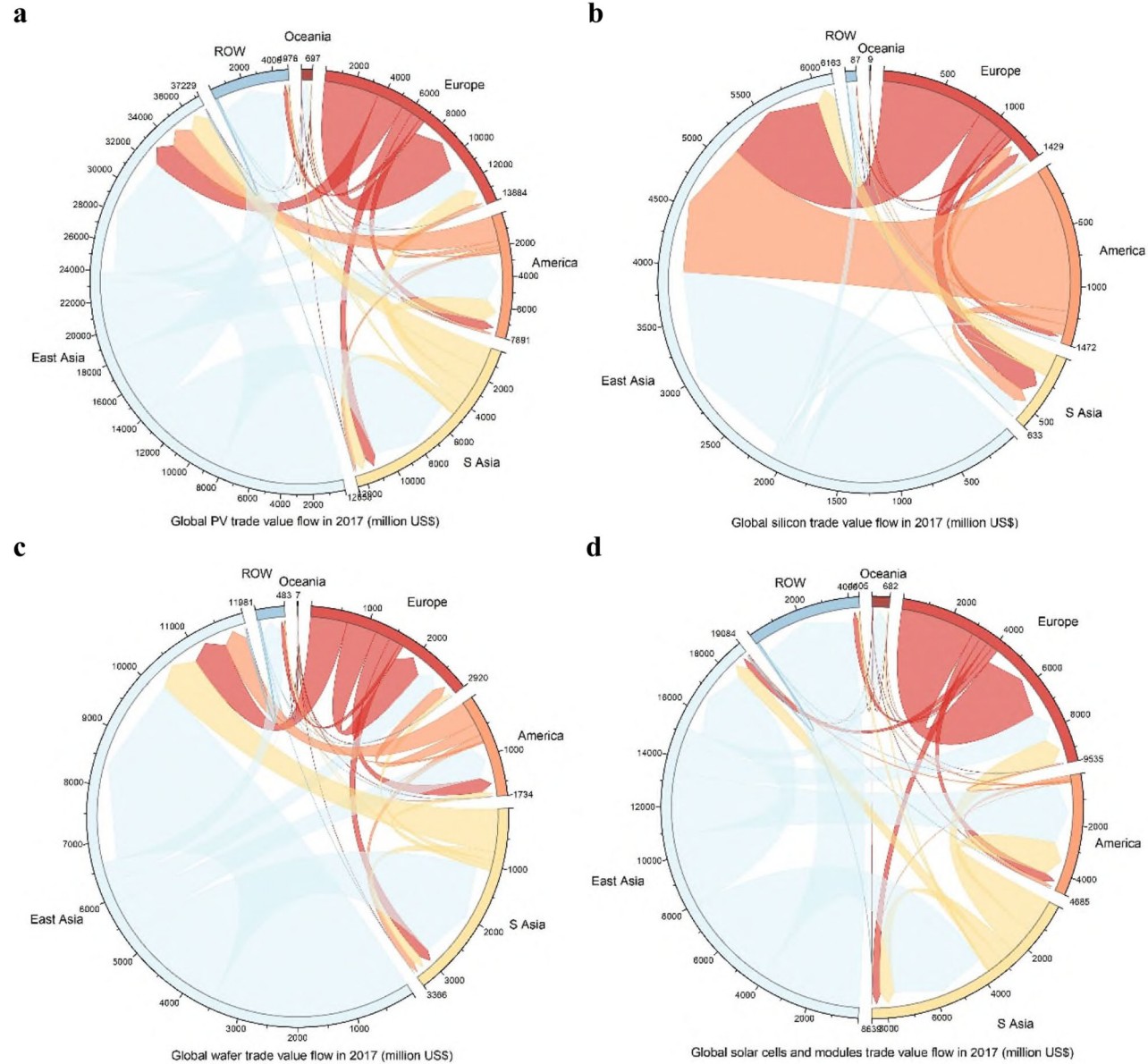

**Fig. 1 Global solar photovoltaic (PV) product trade flow.** Chord graphs present global PV product trade flows in 2017, including total trade value flows and three categories of PV product trade flows. S Asia is the abbreviation for Southeast Asia. ROW is the abbreviation for the rest of the world. Light blue ribbons represent trade flows from East Asia and ROW, and yellow, orange, red and dark red ones are trade flows from Southeast Asia, America, Europe and Oceania, respectively. **a** This figure shows global total PV trade flows. East Asia is the largest PV product exporter and importer region, and Southeast Asia is the second largest importer region. The largest interregional flow is from East Asia to Southeast Asia. **b**–**d** Describe global trade value flows of silicon, wafers, and solar cells and modules, respectively. East Asia leads in all PV product trade. Europe and the Americas tend to produce and export silicon to and import solar cells and modules from Asia. Southeast Asia acts as an important exporter of silicon wafers and solar cells and modules and have close trade links with other regions (see Data sheet 3–5 in Source data for more details). This study converts all currencies to US$ first and then converts them to 2010 US$ for comparability.

Embodied carbon in the PV products and its trade flow are examined. Although a solar PV system generates near zero carbon emissions electricity power during its operation, its life cycle carbon emissions should not be ignored. Global PV product trade is accompanied by embodied carbon transfer, calculated based on global PV product TFM[57] (see the 'Methods' section and Supplementary Information 3 for details). Figure 2 presents the volume and direction of embodied carbon flow (see also Data sheet 6–9 in Source data).

Carbon emissions embodied in the global PV product trade are estimated to be 128.35 million tons of carbon dioxide equivalent (MtCO$_2$e) in 2017, accounting for 0.38% of worldwide fossil fuel

combustion carbon emissions in the same year[58]. In 2017, silicon, silicon wafers, and solar cell and module trade contribute 27.09 MtCO$_2$e, 33.09 MtCO$_2$e and 68.16 MtCO$_2$e, respectively.

East Asia is the largest exporter region of carbon embodied in PV products, at 79.35 MtCO$_2$e in 2017, accounting for 61.83% of the global total. China and Japan are the top 2 carbon outflow countries, aggregately accounting for 43.85% of the global total. East Asia is also the largest embodied carbon importer region, importing 51.55 MtCO$_2$e (40.17% of the global total), followed by Southeast Asia (24.22 MtCO$_2$e, 18.87% of the global total). China and the USA are the top 2 embodied carbon importer countries, aggregately accounting for 25.42% of the global total. The largest

**Table 1 Trade partners identified in PV product trade flow matrix (TFM).**

| Groups of trade partners by region | Country/Economy |
|---|---|
| Oceania | Australia, New Zealand |
| Europe | Austria, Belgium, Bulgaria, Croatia, Cyprus, Czech Republic, Denmark, Estonia, Finland, France, Germany, Greece, Hungary, Ireland, Italy, Latvia, Lithuania, Luxembourg, Malta, Netherlands, Norway, Poland, Portugal, Romania, Russia, Slovakia, Slovenia, Spain, Sweden, United Kingdom |
| Southeast Asia | Brunei, Cambodia, Indonesia, Laos, Malaysia, Myanmar, Philippines, Singapore, Thailand, Vietnam, India, Turkey |
| America | Brazil, Canada, Mexico, USA |
| East Asia | China, Hong Kong, Taiwan, Japan, Republic of Korea |
| ROW | Rest of the world |

India and Turkey are classified into "Southeast Asia" here for convenient illustration. ROW is the abbreviation for the rest of the world.

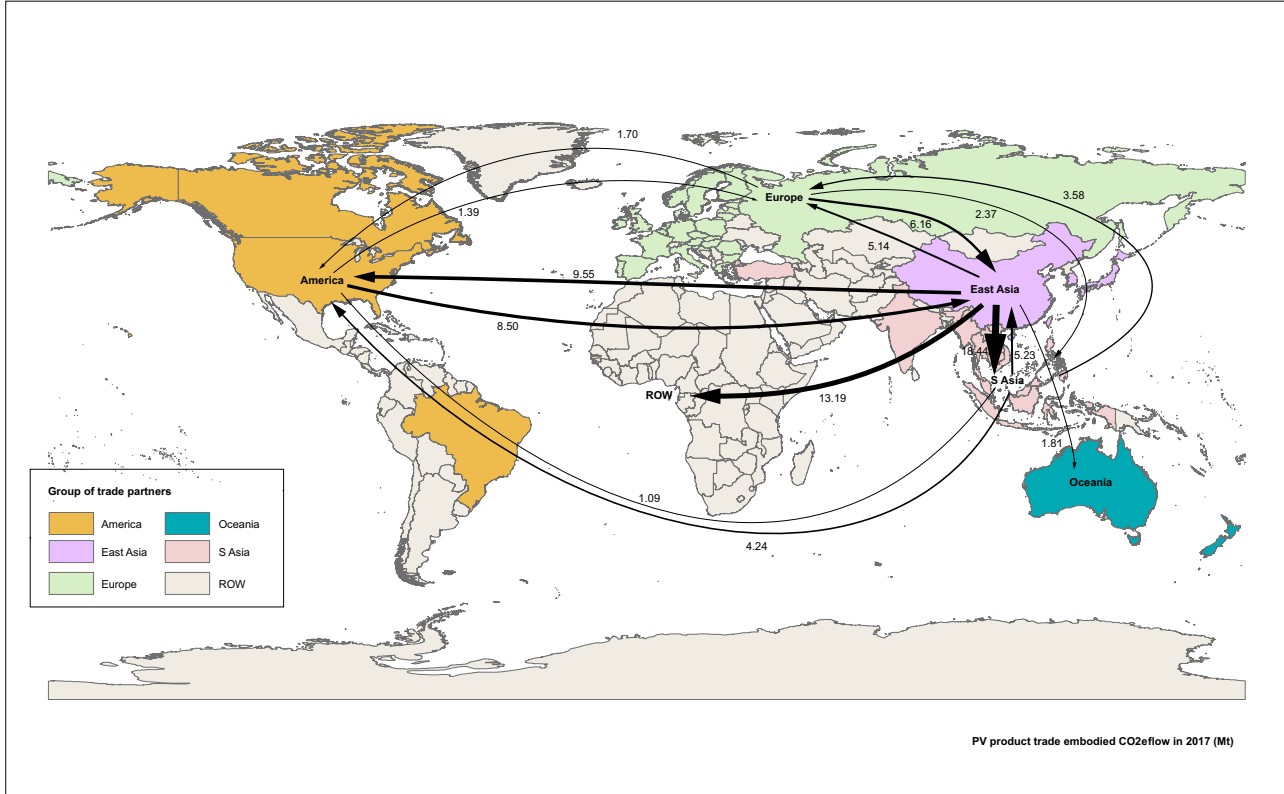

**Fig. 2 Structure and flow of carbon dioxide equivalent (CO₂e) embodied in the global solar photovoltaic (PV) product trade in 2017.** This figure depicts flows of carbon embodied in PV product trade between regions. S Asia is the abbreviation for Southeast Asia. ROW is the abbreviation for the rest of the world. The different coloured areas show various groups of PV trade partners, i.e., purple, pink, green, blue, orange and grey areas represent from East Asia, S Asia, Europe, Oceania, America and ROW, respectively. East Asia is the largest embodied carbon exporter region, with China as the representative. Arrows show embodied carbon flow directions, and the thicknesses of arrows represent the relative sizes of embodied carbon flows (see Data sheet 6 in Source data). The base map is acquired from the open access data source of Resource and Environment Science and Data Centre, Institute of Geographic Sciences and Natural Resources Research, Chinese Academy of Sciences (CAS)[59].

regional embodied carbon flows in the global PV trade are the intra-East Asia flow and the flow from East Asia to Southeast Asia, reaching 31.23 MtCO₂e and 18.44 MtCO₂e, respectively. The largest intercountry flow is from China to India (7.96 MtCO₂e).

At the product level, East Asia is the dominant carbon exporter region in all PV product trade. The leading carbon importer regions in the silicon and silicon wafer trade are East Asia and Southeast Asia, and those in the solar cell and module trade are Europe and Southeast Asia.

In 2017, East Asia and Europe are two net carbon exporter in terms of a positive balance of carbon emissions embodied in trade (BEET$_C$ equalling 27.80 MtCO₂e and 0.96 MtCO₂e, respectively) (see Fig. 3 and the 'Methods' section for more details), and China

is the largest net exporting country, with a BEET$_C$ of 22.91 MtCO₂e, followed by Germany (4.54 MtCO₂e). India is the largest net importing country, with a BEET$_C$ of −8.16 MtCO₂e, followed by Turkey (−2.37 MtCO₂e) (see Data sheet 10 in Source data). Emerging PV markets, such as Australia, Mexico and Brazil, and the USA are also significant net carbon importer countries.

**Status quo carbon net emissions reduction potential of trade-related PV application.** PV power generation helps to reduce carbon emissions when it is used to replace local power generation, especially in grids where carbon-intensive thermal power dominates. This study assumes that imported solar cells and

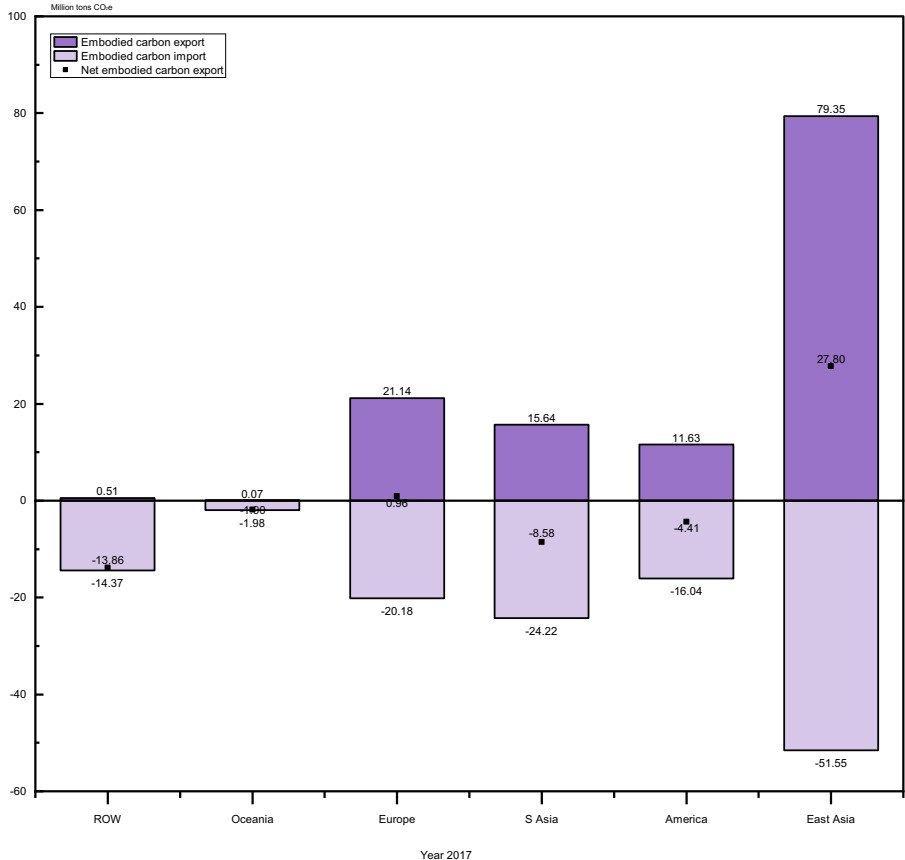

**Fig. 3 Balance of carbon emission embodied in the solar photovoltaic (PV) product trade in 2017.** S Asia is the abbreviation for Southeast Asia. ROW is the abbreviation for the rest of the world. The column chart shows the balance of total carbon emission embodied in the global PV product trade by region: Dark purple bars above the horizontal axis (y = 0) represent carbon emission embodied in exports, and light purple bars under the horizontal axis represent carbon emission embodied in imports. Black dots represent the balance of embodied carbon emission (BEET$_C$.

modules are applied locally to generate PV power and to reduce carbon emissions.

In 2017, the capacity of solar cells and modules globally traded reached 79.65 GW, which would generate 2325.25 TWh during a 30-year lifetime, equivalent to 9.10% of the global electricity generation in 2017[58] (see Supplementary Information 4). East Asia contributes the most PV power generation potential from exported solar cells and modules, accounting for 68.99% of the global total, followed by Europe (15.53%). China is the largest contributing country, accounting for more than 50%. In terms of imports, the most substantial PV power generation potential occurred in Southeast Asia at 670.12 TWh during a 30-year lifetime, and India enjoys the largest PV power generation potential at 435.37 TWh (see Data sheet 11 in Source data).

Based on PV power generation potential, emissions reductions from traded solar cells and modules application can be calculated for two PV power application scenarios, namely substitution for non-PV electricity of the local grid (SSG) and substitution only for fossil fuel combustion or thermal power generation (SST) (see the Methods section for more details). By producing clean solar PV power, the carbon emissions reduction potentials of solar cells and modules traded in 2017 under the SSG and SST scenarios are estimated to be 1218.88 and 1734.59 MtCO$_2$e, respectively, during the 30-year lifetime, equivalent to 3.64% and 5.19% of the global carbon emissions from fossil fuel combustion in the same year, respectively. East Asia contributes the greatest total lifetime carbon abatement potential in the two scenarios, ranging from 906.85 to 1262.24 MtCO$_2$e in 2017 through the export of solar cells and modules, followed by Southeast Asia and Europe. China

contributes 696.87–962.39 MtCO$_2$e of lifetime carbon reduction potential in 2017, accounting for more than 50% of the global total (see Data sheet 12 in Source data).

When carbon emissions embodied in the PV trade and those in the whole PV system, including the balance of system (BOS) and storage system, are considered (see the 'Methods' section and Supplementary Information 3 for more details), net carbon emission reductions from traded solar cells and modules in 2017 are 1053.18 and 1568.89 MtCO$_2$e during the 30-year lifetime under the SSG and SST scenarios, respectively, accounting for 3.15 and 4.69% of global carbon emissions from fossil fuel combustion in the same year[58] (see Table 2). For the net emissions reduction beneficiaries, Southeast Asia receives the largest net emissions reduction potential at 408.84–523.07 MtCO$_2$e. More specifically, India receives a 312.90–384.34 MtCO$_2$e net reduction potential during the 30-year lifetime, becoming the largest beneficiary (see Data sheet 12 in Source data for more detail).

**Projection to power market share and carbon emissions reduction potential of PV application.** This study focuses on the whole world and the 24 major PV application and trading partner countries/economies plus ROW-cpe (see the 'Methods' section for more details), referring to their net-zero emissions or deep decarbonisation strategic pathways or their intended nationally determined contributions (NDCs) and long-term energy mix target commitments (see Supplementary Information 5 for more details), to project their future electricity generation/supply

**Table 2 Lifetime net carbon emission reduction potential from traded solar cells and modules in importing regions (MtCO₂e).**

| Regions | SSG | SST |
|---|---|---|
| Oceania | 50.35 | 63.18 |
| Europe | 119.71 | 215.98 |
| Southeast Asia | 408.84 | 523.07 |
| Americas | 152.51 | 263.39 |
| East Asia | 105.14 | 169.32 |
| ROW | 216.63 | 333.96 |
| Total | 1,053.18 | 1,568.89 |

This table presents the lifetime net carbon emissions reduction potential of traded solar cells and modules over 30-year lifetime in six regions, the SSG scenario represents the substitution of solar photovoltaic (PV) power for local non-PV grid electricity, and the SST scenario represents the substitution of PV power for fossil fuel combustion or thermal power generation. ROW is the abbreviation for the rest of the world.

structure from 2017 to 2060 and to calculate the carbon emissions reduction potential of PV applications, which also serve as the basis for estimating the impacts of trade barriers on the net carbon reduction potential from traded solar cells and modules.

According to our model simulation, the power sector of each country/economy will experience increasing energy demand/supply and substantial energy mix transition, and fossil-fuel electricity in most countries/economies will gradually decrease. Power market share freed up from power-generation facility decommissioning and generated from incremental demand will be mainly occupied by renewable energies, especially solar and wind power generation.

According to our projections, from 2017 to 2060, the global annual electricity supply will increase from 25,441.39 TWh to 51,755.16 TWh. The global annual fossil fuel electricity supply will decrease from 16,595.44 TWh to 6655.18 TWh. The global annual PV power electricity supply will increase from 426.02 TWh to 13,934.96 TWh (with PV power installation capacity increasing from 95.54 GW to 243.25 GW), the proportion of which in the power-generation/supply mix will grow from 1.67% to 26.92% (Fig. 4). These predictive results are close to the World Energy Council's predictions in general[60] (see Data sheet 13 in Source Data). PV application will notably increase, and its proportion in local power supply in most countries/economies will be greater than 10% by 2060. Specifically, countries such as China, the USA and India will witness an obvious shift to renewable electricity supply, and solar power will constitute 21.41%, 21.82% and 46.22% of their power supplies, respectively, by 2060. EU countries, such as the Netherlands, Germany and Italy, will see solar power stand out as an important power source, accounting for more than 24% of each of their power production mixes in 2060.

The present study assumes that solar power is used to substitute for non-PV electricity in the local grid (SSG) or fossil fuel electricity (SST) and calculates the carbon emissions reduction potential of PV applications (see the 'Methods' section for more details). In the SSG scenario, the global annual net carbon reduction potential from global PV power generation will increase from 189.05 MtCO₂e to 1192.57 MtCO₂e. In the SST scenario, it will increase from 273.61 MtCO₂e to 7709.51 MtCO₂e (see Fig. 5, Data sheet 13 in Source data). During the accounting period (2017–2060), global PV utilization will accumulatively bring approximately 51.47 GtCO₂e (SSG) and 182.68 GtCO₂e (SST) net emissions abatements, respectively. In both scenarios, China, India and the USA will see the largest net carbon emissions reduction potentials from PV applications.

**Barriers to the PV product trade would impede global emissions reduction potential.** Freer trade helps to stimulate international trade and production output, reduce PV prices and application costs, and increase PV power capacity and carbon reduction. In contrast, additional trade barriers, including both ordinary customs tariffs and non-tariff measures (NTMs) imposition, will inevitably impede PV trade and production output, raise international PV product prices (see Supplementary Information 6 for more details) and installation costs, and cause PV power capacity and carbon reduction to deteriorate globally.

The long-term (2017–2060) effects of three trade barrier scenarios are simulated and compared with the business-as-usual (BAU) scenario to test the impacts of barriers on solar cell and module trade. The BAU scenario is set based on the status quo tariff rates, antidumping and countervailing duties in global solar cell and module trade in 2017. In the reduced trade barrier scenario (TBS0), half of the ordinary tariffs and NTMs on global solar cells and modules trade are assumed to be removed, representing a freer trade situation. In the foreseeable higher trade barrier scenario (TBS1), increased safeguard duty implemented in the USA–China trade war and trade protection measures undertaken by India in 2018 are reflected. On top of TBS1, the intensified trade barrier scenario (TBS2) is a worsened situation, in which various trade partners impose alleged trade measures on solar cells and modules from all origins (see the 'Methods' section and Data sheet 14 in Source data for more details).

If half of the ongoing trade barriers on solar cell and module are cancelled (TBS0), the global solar cell and module trade volume will increase by 19.97% compared with BAU. Almost all of the major solar cell and module trade partner countries/economies will increase PV imports. These countries/economies will see substantial import increases ranging from 15.88 to 60.68%. Although some countries/economies' domestic production and supply will decrease due to increased imports, their total PV capacity installations will increase by 2.87–14.65%, benefiting from the importing of less expensive foreign products. For major solar cell and module production and exporting countries/economies, the reduction of trade barriers will stimulate their production and export. Considering China as an example, under TBS0, solar cell and module production will increase by 2.15%, and exports will increase by 17.70% compared with BAU, though its domestic application will decrease by 1.90%.

TBS1 and TBS2 will cause all countries/economies to trade fewer solar cells and modules than BAU, and the global total exports/imports will decrease by 2.90% and 6.70%, respectively. In both TBS1 and TBS2, countries/economies imposing extra trade barriers will experience a more apparent decline in solar cell and module imports, and the largest decreases in TBS1 will be in India (9.52%) and the USA (13.93%), and those in TBS2 will be in Europe countries, such as the United Kingdom (24.22%) and France (23.77%), and the import decline in the USA will also be substantial (13.08%). Although domestic solar cell and module producers will benefit from trade protection, due to their relatively poorer competitiveness, their domestic supply will only increase to a small extent and be insufficient to complement the loss of the whole PV power industry from reduced solar cell and module imports[62]. In TBS1, countries/economies imposing higher trade barrier will suffer PV installation decreases, e.g., India by up to 7.34% and the USA by up to 10.15%. In TBS2, countries/economies, including India, the USA, European countries, etc., which exert higher trade barriers, will suffer PV installation decreases of 8.93–16.62%. PV importing countries/economies imposing no extra trade barriers on solar cells and modules will witness slightly increasing imports due to the trade diversion effect and increased PV installation by 0.03–0.69% in TBS1 and by 0.17–1.11% in TBS2. Escalated trade barriers will cause major PV exporters to experience obvious export decreases. For example, solar cell and module exports from China will

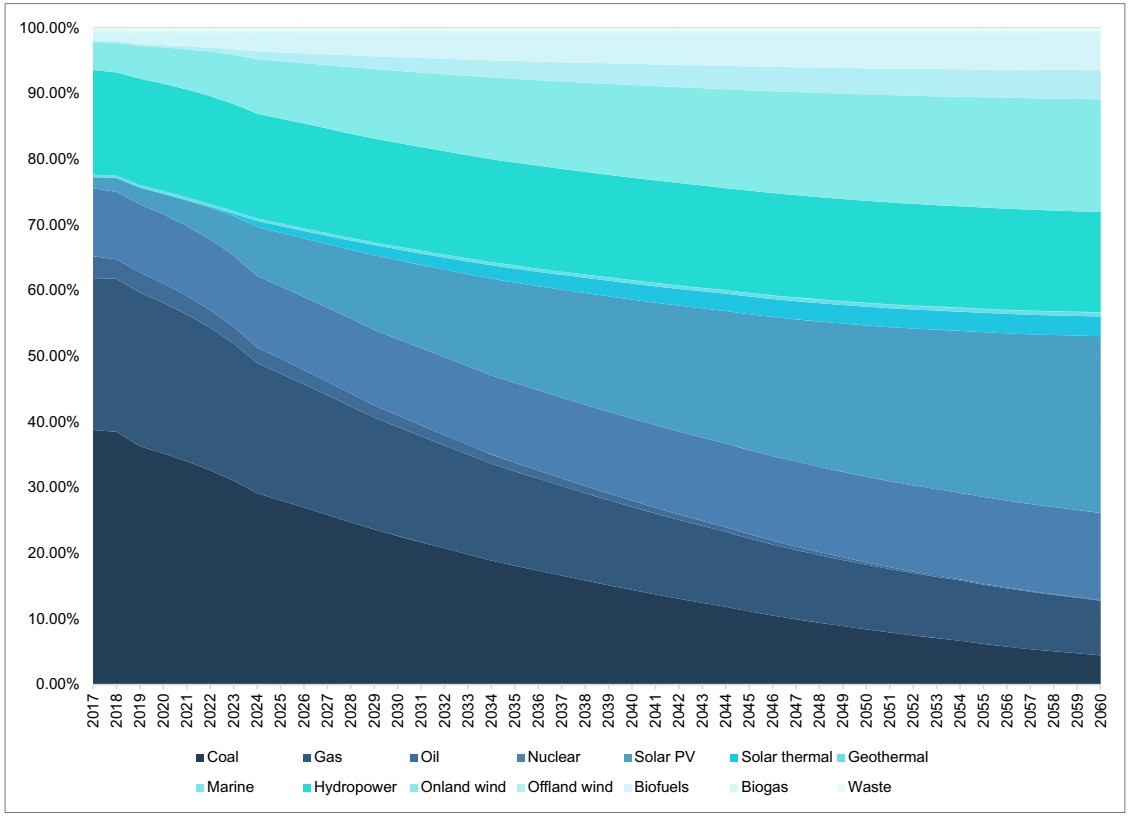

**Fig. 4 Projection to global electricity power production composition from 2017 to 2060.** The ribbons in different colour intensity scales represent the share of various power generation technologies. The world total is obtained from the International Renewable Energy Agency (IRENA)[61]. The power-generation composition is projected by the integrated modelling system (IMS) model simulations based on country-/economy-specific National Determined Contributions (NDCs) or long-term energy mix target commitments of 24 major solar photovoltaic (PV) applications and trade partner countries/economies. That of the countries/economies other than these 24 countries/economies is obtained by subtracting the subtotal of the 24 countries/economies from the world total (See the Methods section, Data sheet 13 in Source data and Supplementary Information 5 for more details).

decrease by 4.60% in TBS1 and 5.63% in TBS2, respectively, compared with BAU. Although escalated trade barriers are conducive to domestic PV installation in PV producing and exporting countries such as the Republic of Korea and China, whose PV installation will increase by 0.55 and 0.53% in TBS1 and by 0.94% and 0.65% in TBS2, the installation increments cannot compensate for installation decreases from global perspective. (See Fig. 6 and Table 3, Data sheet 15–18 in Source data).

In general, compared with BAU, during 2017–2060, global cumulative total solar cell and module production and installation will increase by 7.15% or 752.46 GW, under TBS0, but will decrease by 1.56 and 3.50%, or 164.44 GW and 368.81 GW, under TBS1 and TBS2, respectively (see Fig. 6 and Table 3, Data sheet 15–19 in Source data).

From 2017 to 2060, global PV utilization will cumulatively produce 321,382.70 TWh, 292,687.08 TWh and 291,310.83 TWh under TBS0, TBS1 and TBS2, respectively, indicating that, compared with BAU, global solar power generation will increase by 22,500.60 TWh in TBS0 but decrease by 6195.02 TWh and 7,571.27 TWh in TBS1 and TBS2, respectively (Fig. 7a, see the 'Methods' section and Data sheet 19 in Source data). It is clear that ongoing trade barriers in BAU have restrained the PV product trade and reduced global solar power generation potential, and higher trade barriers (TBS1 and TBS2) will inevitably worsen the loss.

From 2017 to 2060, assuming that solar PV power is used to replace non-PV electricity (SSG) and fossil-fuel electricity (SST), TBS0, TBS1 and TBS2 will lead to global cumulative net carbon

reductions of 55.85–194.88, 49.79–179.39 and 49.83–178.85 $GtCO_2e$, respectively, indicating that status quo trade barrier reduction by half will increase the net carbon reduction potential by 6.26–7.85% or 4.39–12.20 $GtCO_2e$ (SSG-SST), but TBS1 and TBS2 will decrease the global net carbon reduction potential by 1.83–3.35% or 1.67–3.29 $GtCO_2e$ (SSG-SST) and 2.14–3.29% or 1.64–3.83 $GtCO_2e$ (SSG-SST), respectively (see Fig. 7b).

From 2017 to 2060, the freer trade scenario (TBS0) will help to increase carbon emissions reduction in most countries/economies. For example, India and the USA are expected to increase their cumulative net carbon reduction potential by up to 4.85 $GtCO_2e$ and 1.90 $GtCO_2e$ (SST), respectively. The rest countries/economies will see enlarged net carbon reduction potential as well, ranging from 0.01 to 0.53 $GtCO_2e$, while major PV products exporters will see declined net carbon reduction potential, e.g., China will see a decline of 1.35 $GtCO_2e$. Nevertheless, reduction of trade barriers on global PV trade can facilitate worldwide carbon emission reduction.

In the escalated trade barrier scenarios TBS1 and TBS2, countries/economies that exert higher trade barriers will experience larger carbon reduction potential losses; e.g., India and the USA are expected to lose cumulative net carbon reduction potential by 2.48–2.52 and 1.34–1.46 $GtCO_2e$ (SST), respectively. India is expected to suffer even more carbon reduction potential loss than the USA from the restriction on PV goods imports because India had a status quo fossil fuel highly dominated power mix (>80%, while the USA's is approximately 50%). Although trade barriers are not conducive to the reduction of carbon emissions of countries such as India and the United

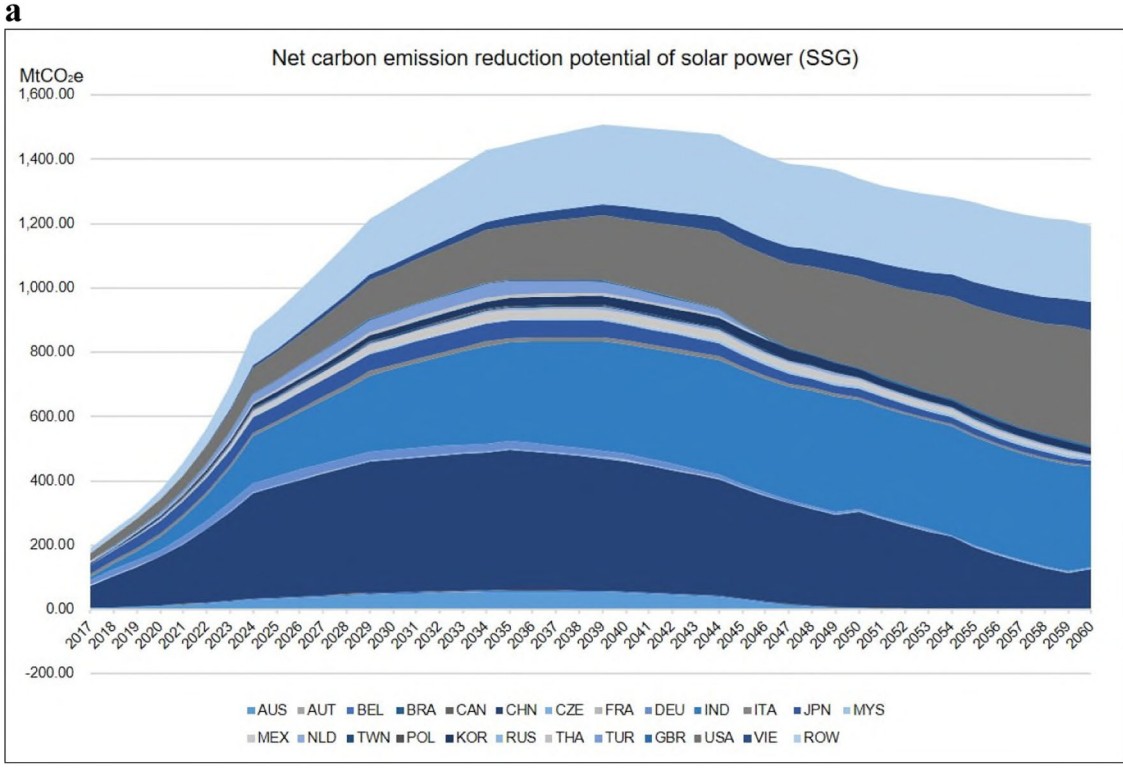

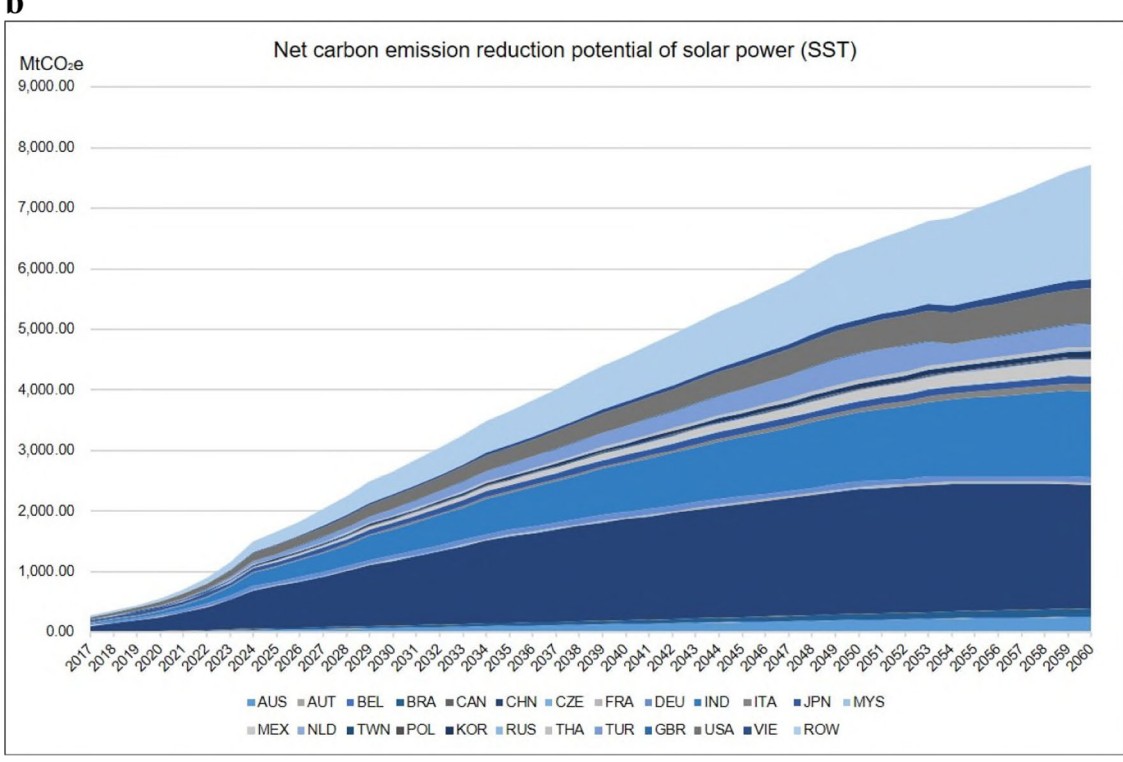

**Fig. 5 Net carbon emissions reduction potential of solar photovoltaic (PV) power from 2017 to 2060.** These figures reflect the net carbon emissions reduction potential of global PV application during 2017–2060. The ribbons in different blue and grey colour intensity scales indicate net carbon emissions reduction potential in different countries/economies. ROW is the abbreviation for the rest of the world. When counting net emission reductions, carbon emissions embodied in solar cells and modules and those embodied in the balance of system (BOS) and storage system are spread evenly over a 30-year lifetime and are then subtracted from the gross carbon reduction potential of PV power replacement. **a** Reflects the net carbon emissions reduction potential when solar PV power is applied to replace non-PV electricity (SSG). The carbon emissions reduction potential in some countries/economies first increases and then decreases. It is reasonable that renewable energy will gradually occupy a larger share in the local grid power mix, and incremental solar PV power substitution will result in smaller carbon abatement. **b** Reflects net carbon emissions reduction potential when solar PV power is applied to replace fossil fuel electricity (SST).

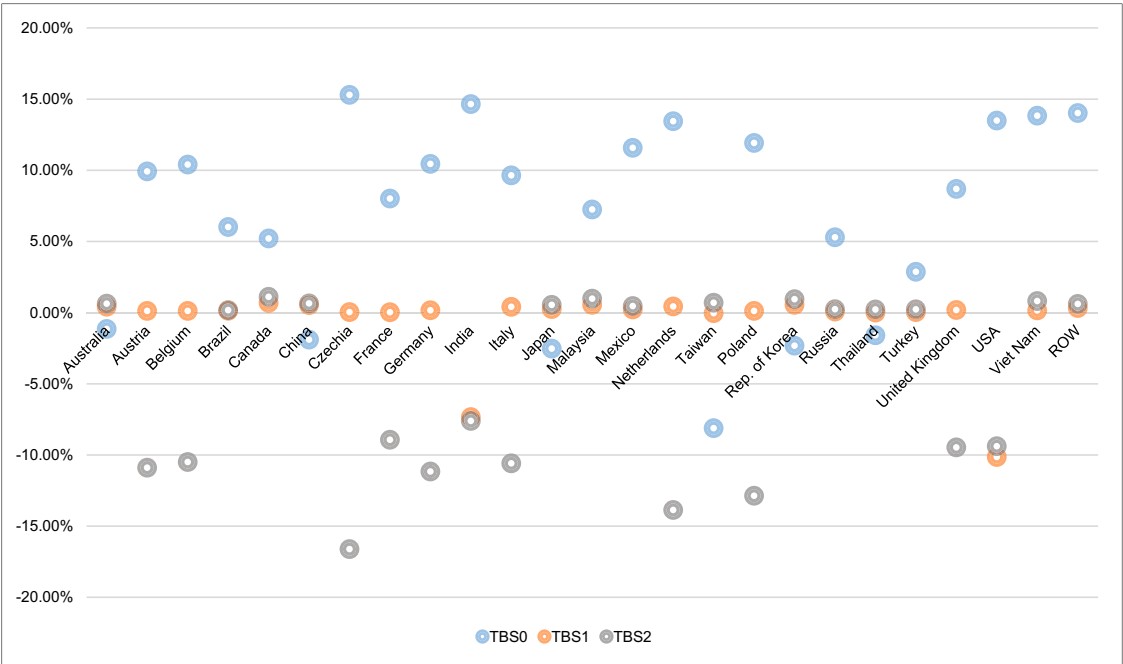

**Fig. 6 Variation rates of solar cell and module application/installation in major trade partner countries/economies under the reduced trade barrier scenario (TBS0), the foreseeable higher trade barrier scenario (TBS1), and the intensified trade barrier scenario (TBS2).** This figure reflects variation rates under TBS0, TBS1 and TBS2. The blue, orange and grey little circles represent variation rates of solar cell and module application/installation in major trade partner countries/economies under TBS0, TBS1 and TBS2, respectively. In TBS0 most countries/economies will see increasing solar photovoltaic (PV) installations. In TBS1 and TBS2, many countries/economies will suffer PV installation decreases due to higher trade barriers. ROW is the abbreviation for the rest of the world.

States which impose higher trade barriers on PV products import, they are conducive to the reduction of carbon emissions of PV products producing and exporting countries such as the Republic of Korea and China, whose cumulative net carbon reduction potential will increase by 0.01–0.02 and 0.37–0.48 $GtCO_2e$ (SST), respectively. However, the overall impacts of trade barrier on PV goods cause the global carbon emission reduction potential to decrease.

## Discussion

The global solar PV product trade plays an important role in facilitating PV product production and utilization and in mitigating climate change. Traded solar cells and modules in 2017 could generate 2325.25 TWh of electricity over their 30-year lifetimes. The environmental benefits are sizable and significant when considering that solar PV power could substitute for local power generation sources. The 30-year lifetime net GHG emissions reduction potential from traded solar cells and modules could be as high as 1053.18 Mt $CO_2e$ (SSG)–1568.89 Mt $CO_2e$ (SST) for those traded in 2017. From 2017 to 2060, global solar power generation will cumulatively contribute 298,882.10 TWh of clean electricity and result in a net reduction in emissions from the BAU scenario of approximately 51.47–182.68 $GtCO_2e$. The present study clarifies that, although the global PV product trade is accompanied by carbon emissions "migration", emissions embodied in the global PV product trade are small compared with the substantial emissions reduction potential that this type of trade can contribute to global public goods by helping to avert climate crises. People can enjoy tremendous net environmental benefits from global PV trade and the related solar power-generation capacity.

Trade protectionism measures not only harm some specific countries/economies but also hurt the whole world by eating into the emissions reduction potential from PV product applications.

Removal of half of the 2017 status quo trade barrier on PV products will increase global cumulative (2017–2060) PV power production by 22,500.60 TWh, leading to an increase in the global cumulative net emissions reduction potential of 4.39–12.20 $GtCO_2e$. Under the extra trade barrier imposing scenario, such as additional trade barriers on PV products associated with the USA–China trade war, from 2017 to 2060, global cumulative PV power production will be 6195.02–7571.27 TWh less under TBS1 and TBS2 than in the BAU scenario, causing a decline in the global cumulative net emissions reduction potential of 1.67–3.29 $GtCO_2e$ and 1.64–3.83 $GtCO_2e$, respectively. Therefore, there are strong environmental cases for facilitating the trade of PV product trade globally.

There are also several policies and measures that could support the recommended facilitation of the PV product trade. The most straightforward of these measures is lowering trade barriers, including ordinary tariffs and NTMs; however, this change is very unlikely to happen without coordination from regional and global institutions. A concrete channel through which this coordination can occur involves incorporating PV products into the 'environmental goods list' of the WTO and regional free trade agreements and excluding these products from protectionism and trade barriers. There is also a case for strengthening coordination between regional and global trade organizations and the United Nations Framework Convention on Climate Change (UNFCCC). For example, the UNFCCC could encourage countries to include the global impacts of PV product trade in their NDCs, which countries pledged as part of the Paris Agreement. Finally, these efforts would be given additional support with enhanced coordination and information sharing across agencies responsible for trade portfolios and climate change. Overall, the analysis of this article could provide a firm evidence base for collaboration at multiple decision levels, expanding the global solar PV product market and carbon emissions reductions.

**Table 3 Changes in solar cell and module exports, imports, outputs and application under TBS0, TBS1 and TBS2 in contrast to BAU scenario (%).**

| Country/Economy | Change in export | | | Change in import | | | Change in output | | | Change in application/installation | | |
|---|---|---|---|---|---|---|---|---|---|---|---|---|
| | TBS0 | TBS1 | TBS2 | TBS0 | TBS1 | TBS2 | TBS0 | TBS1 | TBS2 | TBS0 | TBS1 | TBS2 |
| AUS | 30.33 | −2.04 | −3.72 | 0.40 | −0.14 | −0.22 | 12.14 | −1.03 | −1.82 | −1.15 | 0.41 | 0.63 |
| AUT | 22.29 | −0.47 | −10.44 | 25.53 | −0.04 | −21.60 | 5.20 | −0.28 | 0.63 | 9.92 | 0.12 | −10.90 |
| BEL | 27.86 | −0.39 | −21.09 | 25.15 | −0.04 | −21.45 | 2.24 | −0.22 | 0.16 | 10.41 | 0.12 | −10.49 |
| BRA | 59.26 | −0.50 | −1.11 | 31.93 | 0.11 | 0.14 | −17.96 | −0.38 | −0.52 | 6.02 | 0.13 | 0.17 |
| CAN | 26.05 | −5.92 | −8.36 | 30.10 | −0.45 | −0.59 | 5.28 | −2.83 | −4.13 | 5.21 | 0.69 | 1.11 |
| CHN | 18.69 | −5.09 | −6.26 | 60.12 | −0.78 | 0.03 | 2.15 | −1.26 | −1.64 | −1.90 | 0.53 | 0.65 |
| CZE | 23.34 | −0.28 | −13.68 | 22.14 | −0.01 | −17.96 | 15.51 | −0.20 | −9.76 | 15.30 | 0.04 | −16.62 |
| FRA | 29.39 | −0.60 | −13.78 | 29.81 | −0.06 | −25.51 | 3.01 | −0.22 | 2.71 | 8.01 | 0.03 | −8.93 |
| DEU | 27.13 | −0.68 | −12.11 | 27.91 | −0.04 | −23.33 | 9.53 | −0.41 | −2.16 | 10.44 | 0.17 | −11.16 |
| IND | 31.33 | −9.00 | −14.96 | 20.69 | −9.99 | −10.65 | −10.59 | 6.51 | 5.91 | 14.65 | −7.34 | −7.61 |
| ITA | 24.73 | −5.29 | −11.09 | 25.48 | −0.24 | −21.99 | 8.99 | −2.96 | −1.90 | 9.64 | 0.41 | −10.59 |
| JPN | 52.69 | −0.67 | −1.85 | 1.29 | 0.31 | 0.63 | 8.31 | −0.31 | −0.72 | −2.53 | 0.25 | 0.55 |
| MYS | 19.48 | −4.12 | −6.88 | 25.88 | −0.30 | −0.52 | 10.33 | −2.70 | −4.54 | 7.24 | 0.54 | 0.98 |
| MEX | 37.26 | −1.10 | −26.59 | 21.10 | −0.09 | −0.22 | −19.38 | −0.42 | −1.46 | 11.57 | 0.22 | 0.46 |
| NLD | 26.16 | −0.25 | −16.67 | 25.55 | −0.07 | −21.39 | 13.13 | −0.22 | −8.23 | 13.45 | 0.43 | −13.87 |
| TWN | 34.72 | 0.08 | −4.35 | 66.20 | 0.29 | −0.72 | 15.76 | 0.03 | −2.41 | −8.12 | −0.04 | 0.69 |
| POL | 28.15 | −0.69 | −16.09 | 23.81 | −0.04 | −20.63 | 3.65 | −0.35 | 1.59 | 11.92 | 0.12 | −12.87 |
| KOR | 19.59 | −4.34 | −7.58 | 2.99 | −0.70 | −1.26 | 11.56 | −2.68 | −4.70 | −2.32 | 0.55 | 0.94 |
| RUS | 36.42 | 4.79 | −4.94 | 30.70 | 0.15 | 0.12 | −13.69 | −0.15 | −0.67 | 5.29 | 0.07 | 0.24 |
| THA | 28.85 | 0.16 | −3.23 | 47.93 | 0.28 | 0.41 | −2.60 | −0.09 | −0.89 | −1.59 | −0.01 | 0.22 |
| TUR | 32.72 | −0.60 | −1.78 | 36.68 | 0.11 | 0.61 | −13.19 | −0.19 | −0.91 | 2.87 | 0.04 | 0.23 |
| GBR | 34.69 | −1.80 | −16.72 | 31.01 | −0.06 | −26.07 | −0.67 | −0.57 | 5.50 | 8.69 | 0.18 | −9.47 |
| USA | 32.92 | −12.61 | −27.56 | 23.60 | −14.69 | −14.71 | −14.69 | 11.98 | 10.13 | 13.49 | −10.15 | −9.39 |
| VIE | 24.21 | −2.90 | −9.99 | 22.10 | −0.08 | −0.37 | 15.34 | −2.00 | −7.02 | 13.84 | 0.15 | 0.81 |
| ROW$_{CPE}$ | 20.47 | −0.91 | −6.01 | 17.10 | −0.11 | −0.24 | 10.37 | −0.67 | −4.16 | 14.02 | 0.33 | 0.61 |

This table shows changes in solar photovoltaic (PV) trade export value, import value and output of each trade partner and major countries/economies under the reduced trade barrier scenario (TBS0), the foreseeable higher trade barrier scenario (TBS1), and the intensified trade barrier scenario (TBS2). ROW$_{CPE}$ includes countries/economies other than the 24 largest PV trade partners identified in the GSIM 6.0 model and were treated as one economy. This study converted all of the currencies to US$ first and then converted them to 2010 US$ for comparability.

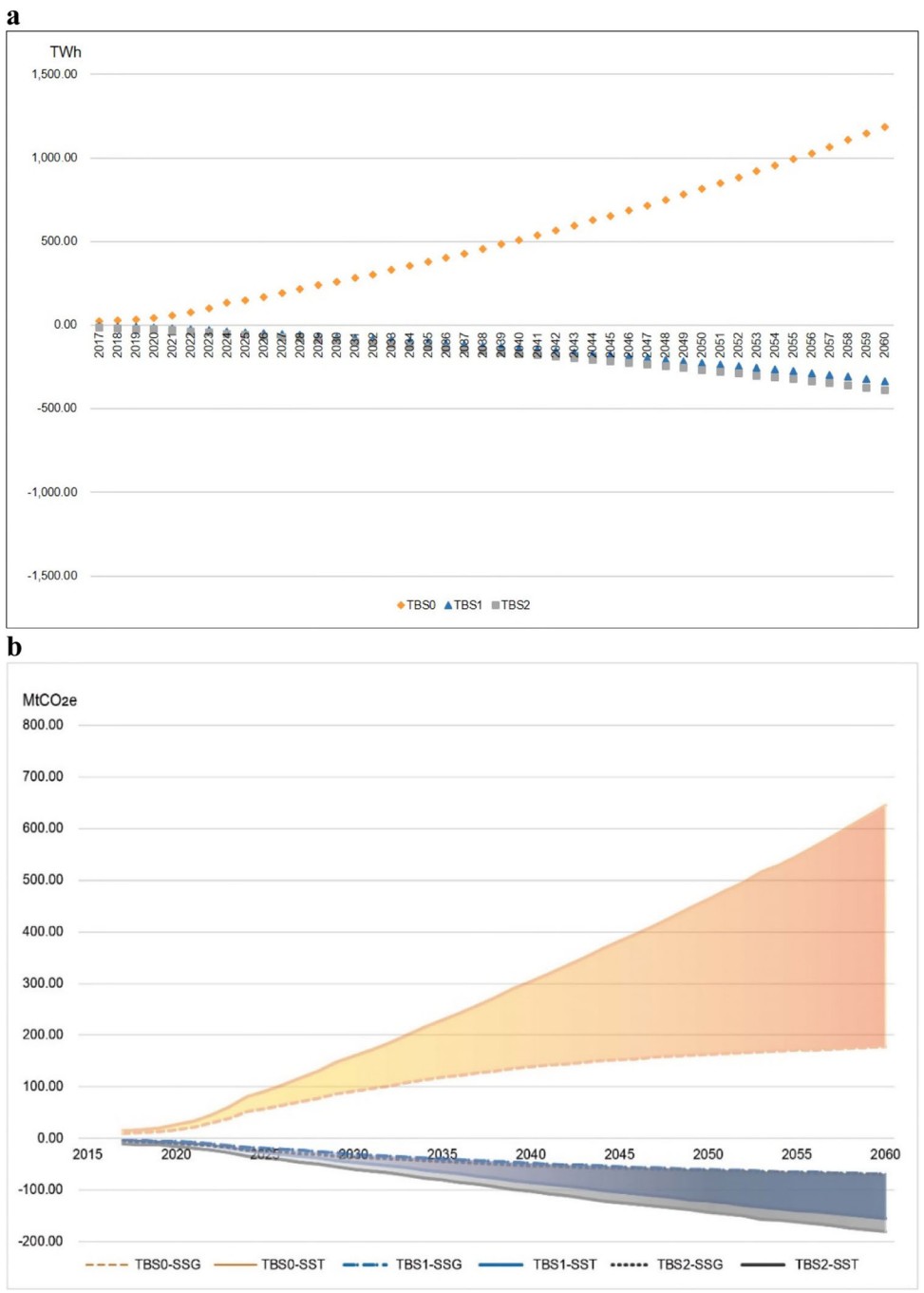

**Fig. 7 Projected variations in global annual solar power generation and carbon emissions reduction potential in different tariff scenarios. a** reflects the projected variations in global annual solar power generation during 2017–2060 under TBS0, TBS1 and TBS2. The orange, blue and grey dotted lines correspond to the reduced trade barrier scenario (TBS0), the foreseeable higher trade barrier scenario (TBS1), and the intensified trade barrier scenario (TBS2), respectively. **b** reflects the variation in the global net carbon emissions reduction potential of solar PV power application during 2017–2060 under TBS0, TBS1 and TBS2, assuming that solar PV power will be used to replace local electricity. The orange, blue and grey colours correspond to TBS0, TBS1 and TBS2, respectively. Dotted dashed lines reflect carbon emission reduction variation when solar PV power is used to replace non-PV electricity (SSG). Solid lines reflect carbon emission reduction variation when solar PV power is used to replace fossil fuel thermal power electricity (SST). The ribbons between dotted dashed lines and solid lines in corresponding scenarios represent the ranges of carbon emissions reduction potential estimations between SSG and SST.

## Methods

The present study considers and refers to the status quo PV technology composition/mix of mono-crystalline silicon (35%), multi-crystalline silicon (61%) and thin-film (4%) solar cells and modules in the PV market, or refers mainly to crystalline-based PV technologies, to perform the analysis. The four major products in the global PV industry chain are silicon, silicon wafers, solar cells, and PV modules[5]. In the present study, solar cells and modules are treated as a single set of product segments for convenience. Because international statistics on PV product production and trade in the early years were sparse, and 2017 set the historical records for 100 GW of global newly installed PV power capacity, we choose 2017 as the base year (status quo) to obtain the global snapshot observations and then project to 2060. First, a trade flow matrix (TFM) is constructed to describe the global PV product trade situation. Second, the embodied carbon in each product's trade, as well as the BOS and batteries are calculated. Third, the net carbon emissions reduction potential of PV power generation is predicted through 2060 with a technology-based dynamic model, integrated modelling system (IMS).

Finally, a computable partial equilibrium model, GSIM, is applied to simulate the impacts of removal and the exerting of trade barriers on the PV product trade, and the resulting gains and losses in carbon emissions reduction potential are calculated (see Supplementary Fig. 1 for a technical route map).

### Multilateral PV product trade and trade flow matrix (TFM) construction.
Researches on international trade and environmental issues usually employ the multiregional input-output (MRIO) method and data[63–69]. Global MRIO databases, such as GTAP, WIOD, and Eora, are based on macroeconomic sector data, from which information about PV products and their global trade is difficult to extract. Existing studies have mainly referred only to trade statistics for solar modules to account for embodied carbon flow and have covered only limited countries/regions[9,38,70].

To form a global PV product trade data set, this study choose the major PV product trade partners, the cumulative share of which exceeded 80% for each product based on the ABRAMS world trade wiki[71]. Then, country/economy-specific PV product trade data were extracted from multilateral institutional databases, e.g., the UN Comtrade database[72], and databases of customs and trade departments from dozens of countries/economies[72–88] to construct PV product TFMs, which serve as the basis for conducting calculations and analyses of the global PV product trade.

Harmonized system (HS) commodity codes were used to retrieve trade data for PV products (HS6 code, silicon-280461, silicon wafer-381800, and solar cells and modules-854140). More specific trade data with 8 or 10 digits were obtained from official statistics or customs of various countries/economies. Fifty-three countries/economies and the "rest of the world" (ROW) are included in the TFM, and they are divided into six groups/regions, namely, Oceania, Europe, Southeast Asia (ASEAN plus India and Turkey), Americas, East Asia, and ROW (see Table 1). ROW refers to all countries/economies that are not explicitly identified, treated as one group of trade partners.

We built TFM $A_{ij}$ for each PV product category, where $i$ denotes the exporting country/economy, and $j$ denotes the importing country/economy. $a_{ij}$, an element of $A_{ij}$, represents the trade value of each product exported from $i$ to $j$.

The trade values between ROW and other identified countries/economies were not directly known owing to limited information, so we estimated the export value from ROW to country $j$ based on an average level of fifty-three countries/economies (see Supplementary Information 2).

The trade value statistics obtained from various sources are in the current prices of various currencies. This study converts all of the currencies to US$ first and then converts them to 2010 US$ for comparability.

Existing PV product trade studies have mostly been conducted simply based on the UN Comtrade database, and few of these studies have noted re-export and re-import problems in the database[38,70]. Proportions of re-export to total export value in some economies are quite considerable; for instance, in 2017, more than 99% of the total value of solar cells and modules exported from Hong Kong SAR were attributed to re-export, and in Canada and the USA, the proportions were both greater than 11%, which essentially could distort real global PV trade volume. In this study, the export data were traced back and assigned to the actual sources (exporters) and destinations (importers) to construct an adjusted TFM (see Supplementary Table 1, Supplementary Information 2).

### Accounting embodied carbon emissions in PV products and their global trade.
To estimate the net carbon emissions reduction contribution from traded PV products (mainly considering solar cells and modules), it is necessary to account for embodied carbon emissions in the PV trade. PV power generation is relatively clean and produces near zero emissions, but certain amounts of GHGs are emitted during the PV production process[10,22,64,66].

To calculate carbon emissions embodied in PV products and their trade, life cycle carbon emissions coefficients of PV products are obtained from the Ecoinvent database[57] (see Supplementary Table 2). Currently, silicon solar cells and modules dominate the global PV industry, with thin-film solar cells and modules accounting for only a small market share. This study, based on the PV market share situation in 2017[89], calculates the comprehensive carbon emission coefficients of PV products to perform embodied carbon emissions accounting. Life cycle emissions of PV products would reasonably decrease with PV production technology improvement. Based on LCA results from Ecoinvent, this study considers improved technologies, including reduced silicon wafer thickness, reduced material usage (such as silicon, silver content in the metallization paste, copper) and kerf losses, wafer size, solar cell and module power, electricity consumption reductions, potential changes in the electricity grid mix, etc., to calibrate the emissions coefficients for PV products (see Supplementary Table 3). Since the PV product trade volume in the constructed TFMs are in US$ but not in physical quantities, the emissions coefficients measured by PV product quantity are converted into emissions per unit value in US$ based on the prices of PV products. The Wind database[90] is used to obtain the international average prices of PV products in 2017. Life cycle carbon emissions coefficients for PV products are listed in Supplementary Information 3 and Supplementary Tables 2–3.

Based on PV product TFMs, embodied carbon can be calculated by Eq. (1):

$$EC_{i,j} = EF_{i,C} \times a_{i,j} \qquad (1)$$

where $EC_{ij}$ is the embodied carbon transferred from country/economy $i$ to country/economy $j$. $EF_{i,C}$ is the carbon emission coefficient of the PV product in country/economy $i$.

The balance of emissions embodied in trade ($BEET$)[91,92] can be calculated by Eq. (2) to reflect the net carbon contribution from PV trade for a specific country/economy:

$$BEET_{Ci} = \sum_j EC_{i,j} - \sum_j EC_{j,i} \qquad (2)$$

where $BEET_{Ci}$ is the balance of the embodied carbon flows of country/economy $i$ in its PV goods trade with other countries/economies.

### Accounting emissions reduction potential generated from traded solar cells and modules.
To calculate the emissions reduction potential of traded PV products, several steps are involved. First, trade values of solar cells and modules are converted into trade volumes based on product prices. Second, solar power generation potential is predicted for the installed capacity attributed to solar cell and module trade. Third, the lifetime net emissions reduction potential is determined based on the PV power generation potential and carbon embodied in the PV system.

Two scenarios are assumed for the emissions reduction potential estimation. The first scenario is substituting PV power for non-PV electricity in the local grid (SSG), and the emissions reduction potential is estimated with the average emissions coefficients of the local grid (a mix of various energy sources, excluding PV power); the second scenario is substituting PV power for thermal power generation (SST), and the emissions reduction potential is estimated with the average emissions coefficients of local fossil fuel combustion or thermal power plants (see Data sheet 20–21 in Source data).

For PV product $m$, the trade volume of export country/economy $i$ to import country/economy $j$, $TV_{ij}$, is calculated by Eq. (3):

$$TV_{i,j} = a_{i,j} \div P_m \qquad (3)$$

where $P_m$ is the international price of solar cells and modules, measured in US$ per kWp $P_{peak}$ ($P_{peak}$ is the nominal rated maximum power in kWp of the system based on 1 kW/m² radiation at standard test conditions (STC), which is associated with the PV module power conversion efficiency). For the 2017 status quo, the corresponding conversion efficiencies are assumed to be 16–25% for mono-crystalline and 14–18% for multi-crystalline, respectively[5].

The PV power generation potential in country $j$ resulting from solar cells and modules imported from country $i$, $QE_{i,j}$, is obtained by Eq. (4)[93] (see Supplementary Fig. 4):

$$QE_{ij} = TV_{i,j} \times \frac{Irra_{i,j}}{H_{stc}} \times PR \times SL \times LR \qquad (4)$$

$$LR = \sum_{y=1}^{SL} \frac{(100 - 0.7 \times (SL - 1))}{100} \qquad (5)$$

where $Irra_j$ is the annual irradiation (kWh/m²/year) in country $j$, and national irradiance data are obtained from the IEA-PVPS[94] and PVGIS databases[95]. $H_{stc}$ is the irradiance at standard test conditions, equal to 1 kW/m². $PR$ is the performance ratio, which is set to 80%. $SL$ is the PV system lifetime ranging between 20 and 40 years, and it is set to 30 years here. $LR$ is the coefficient considering the total efficiency loss ratio (0.7% per year) of the PV system. $PR$, $SL$ and $LR$ are set based on the IEA recommended value[96].

This study defines the emissions reduction factors of $CO_2$ attributed to PV application by Eq. (6):

$$ERF_{i,C} = EF_{i,C,LNPV} \text{ or } EF_{i,C,Ther} \qquad (6)$$

where $ERF_{i,C}$ are the carbon emissions reduction factors of PV application in country/economy $i$. $EF_{i,C,LNPV}$ represents the carbon emissions coefficient of local non-PV electricity in country/economy $i$, which is used in the SSG scenario. $EF_{i,C,Ther}$ is the carbon emission coefficient of local fossil fuel combustion or thermal power generation, which is used in the SST scenario.

For the SSG scenario, the $EF_{i,C,LNPV}$ factors of EU countries, the USA, Canada, Norway, Turkey, Australia, New Zealand and China are calculated based on official statistics[97–105], those of other identified trade partner countries/economies are calculated according to IEA statistics[6,106], and the global average emission factor obtained from IEA[107] is applied to ROW.

For the SST scenario, the $EF_{i,C,Ther}$ data of the USA, Canada, Australia, EU countries, Norway, Turkey, other OECD members and China are obtained from official statistics[6,98–101,103,104,108–118]; coal-fired power generation data are obtained for Australia; and thermal power generation data are obtained[100,114] for the remaining countries/economies. For countries/economies with thermal power generation information that could not be directly obtained, we estimate their $EF_{i,C,Ther}$ based on the EDGAR database[119] and local electricity structures[6]. Since the EDGAR database only provides emissions data back to 2012, we then extrapolate it to 2017 according to technology progress in the countries/economies' electric power structures[120,121].

The quantity of carbon emissions reduction is calculated by Eq. (7).

$$QER_{i,j,C} = QE_{i,j} \times ERF_{j,C} \qquad (7)$$

where $QER_{i,j,C}$ is the carbon emissions reduction potential of the application of solar cells and modules exported from country/economy $i$ to $j$ and generating power in $j$.

Because production of the BOS and storage system also leads to carbon emissions, this study calculates the carbon emissions embodied in the two parts based on emission coefficients from the latest studies (see Supplementary Table 4)[26,27]. Carbon emissions embodied in traded solar cells and modules, BOS and storage systems are deducted from the emissions reduction potential of traded solar cell and module applications to obtain the net carbon reduction potential (see Data sheet 12 in Source data).

**A technology-based integrated dynamic model to project power market share.** This study adopts the integrated modelling system (IMS) model to simulate and depict the future energy structure of power production and supply in different economies. IMS, as a powerful bottom-up dynamic energy-economy-environment (3E) model, was originally developed by the Energy and Materials Research Group (EMRG) at Simon Fraser University (SFU) tailored for Canada known as CIMS, and reformed to adapt to multiple countries/economies by the present study. It has advanced in representing technology composition and characteristics in specific sectors with a dendritic structure, capturing technical parameters, such as capital cost, operation and maintenance costs, life spans, fuel consumption, and emissions coefficients. It is capable of explicitly describing technology competition and substitution for various industries, assuming that old technologies exit the market and new technologies compete for vacant and grown market share. The ratio of the life-cycle cost (LCC) of a technology to the total LCC of all technologies in the power market is the decisive factor during the market share allocation process. The simulation principles of IMS facilitate their applicability on a broad temporal and spatial scale[122–124] and in various countries/economies and sectors[125,126].

In this study, IMS is employed to simulate the power sector evolution in various countries/economies from 2017 to 2060, with an emphasis on the competition of solar PV with other power generation technologies, including coal, gas, oil, nuclear, solar thermal, geothermal, marine, hydropower, onshore wind, offshore wind, biofuel, biogas and waste incineration, for 24 important PV trade partner countries/economies plus ROW-cpe (see Supplementary Information 5). Technology competitions are simulated according to various constraints, such as total power demand, energy resource constraints, and national carbon emissions reduction targets, which are acquired from national development plans, energy development outlooks, NDC target commitments, long-term energy strategic plans and decarbonisation pathway research reports of various countries/economies[61,127–150] (see Supplementary Information 5 and Supplementary Table 5). Carbon emissions coefficients and the power generation cost of each power generation technology are drawn from TIMES (The Integrated MARKAL and EFOM System) guidelines to represent country-/economy-specific characteristics[151]. The energy consumption factors of various power generation technologies are obtained from relevant researches and official statistics[152–155]. The IMS model provides power generation/supply market structure dynamic simulation results with a 5-year interval, so this study uses interpolation to acquire the power production composition of each year in each country/economy (see Supplementary Fig. 5). These results of IMS model simulation are compared with those of TIMES model (see Supplementary Fig. 6).To predict the carbon emissions reduction potential of PV applications, this study calculates carbon emissions coefficients for non-PV power generation mix and fossil fuel power generation in various countries/economies using Eqs. (8) and (9). The carbon emissions reduction potential of solar power substitution for the non-PV power mix and fossil fuel power are then determined with Eqs. (10) and (11):

$$EF_{i,C,nonPV,y} = \frac{\sum_k (G_{k,y} \times \varepsilon_k)}{\sum_{k \neq PVpower} G_{k,y}} \qquad (8)$$

$$EF_{i,C,fossil,y} = \frac{\sum_k (G_{k,y} \times \varepsilon_k)}{\sum_{k=coal,gas,oil} G_{k,y}} \qquad (9)$$

$$QER_{i,C,non-PV,y} = EF_{i,C,nonPV,y} \times G_{k=PVpower,y} \qquad (10)$$

$$QER_{i,C,fossil,y} = EF_{i,C,fossil,y} \times G_{k=PVpower,y} \qquad (11)$$

$EF_{i,C,nonPV,y}$ and $EF_{i,C,fossil,y}$ represent the carbon emissions coefficients of non-PV power mix and fossil-fuel combustion power generations, respectively, in country/economy $i$ and year $y$. $EF_{i,C,nonPV,y}$ and $EF_{i,C,fossil,y}$ also serve as the emissions reduction factors of PV application. $QER_{i,C,nonPV,y}$ and $QER_{i,C,fossil,y}$ represent the carbon emissions reduction potential of PV application in country/economy $i$ and year $y$. $G_{k,y}$ is the power generation of technology $k$ in year $y$ obtained from the IMS. $\varepsilon_k$ is the carbon emissions factor of technology $k$. Similarly, production-related $CO_2$ emissions of solar cells and modules, BOS and storage systems are considered, which are deducted from $QER_{i,C,nonPV,y}$ and $QER_{i,C,fossil,y}$ to obtain the net carbon reduction potential.

**Trade barrier scenarios and computable partial equilibrium model simulation.** Trade barriers consist of ordinary tariffs and non-tariff measures (NTMs)[156]. According to the WTO[157] and MOFCOM[158], countries/economies, such as Brazil imposed tariffs on PV products. NTMs are those such as antidumping measures, countervailing measures, or safeguard measures, which are more frequently applied in the course of trade frictions or conflicts and continually influence global PV product trade, production and application[5]. Several major PV product trade conflicts occurred in 2017; for example, the USA complained that the domestic PV industry was damaged by imported products and vowed to impose additional safeguard measures (investigation and extra duty included) on imported solar cells and modules, which led to objections from China, the Republic of Korea, Mexico. The European Union restricted the minimum import price (MIP) and maximum shipping volume of Chinese PV products. Turkey and India also imposed tariffs on imported solar modules from China[5]. On April 16, 2020, the USTR removed the Section 201 tariff exclusion for the bifacial solar module[40]. Based on the above facts, three trade policy scenarios and business-as-usual (BAU) scenarios are assumed (see Supplementary Information 6 and Supplementary Table 6 for detailed scenario settings).

The global simulation model (GSIM), which is a multiregional computable partial equilibrium model, was developed and expanded by Francois and Hall[159], and it focuses on industry analysis from a global perspective, allowing for rapid and relatively transparent analysis of trade policy issues with minimal data and computational requirements[56,160–162]. This study adopts GSIM 6.0 to simulate the impacts of trade barriers on global solar cell and module prices (see Supplementary Fig. 7), trade (import and export), production, and application/installation. Based on the global PV trade structure indicated by the 2017 TFM, the GSIM simulation results of PV application changes in various countries/economies are fed back into the IMS model to predict the changes in solar power generation and the corresponding emissions reduction potential from 2017–2060.

The GSIM 6.0 model can incorporate a maximum of 25 countries/economies or trade partners, so the 24 largest PV product trade partners (which are also the major PV producers and users) and ROW-CPE (composed of the countries/economies other than the largest 24) are included (see Supplementary Table 7). Solar cells and modules, as the major traded PV products and key objectives of trade conflicts, are used to perform the simulations. Trade value data are obtained from UN Comtrade[72], tariff rates are obtained from the WTO[157] and MOFCOM[158] (Data sheet 14 in Source data), ad-valorem equivalents (AVEs) of NTMs relative to solar cells and modules in 2015, which is corresponding to HS code 854140, are obtained from the latest research[163]. Supply, demand and substitution elasticities of solar cell and module trade are obtained from current research, which are econometrically estimated[56] (see Supplementary Table 8). Solar cells and modules production and its changes in various countries/economies are estimated based on local PV production or production capacity statistics. Domestic applications of solar cells and modules are obtained by deducting net exports (export minus import) from domestic production. Detailed data sets of PV production, trade, and application/installation in major trade partner countries/economies are shown in Supplementary Information 6.

**Uncertainty and sensitivity analysis.** Considering the complexity of the analysis process, there could be various uncertainties originating in the present study, for example, the technology improvement which will cause decrease of life cycle emissions coefficients, which will influence balance of carbon embodied in the PV product trade (see Supplementary Fig. 2), and increase of conversion efficiency can enlarge the net carbon emission potential of traded PV products (see Supplementary Fig. 3), the projection results of the long-term power production & supply mix in various countries/economies with IMS and other models can be different, and the PV trade and consumption prediction with GSIM model can also be affected by varied tariff rates, NTMs levels and substitution elasticity settings, etc. Thus, sensitivity analysis for these key analytical steps are conducted, present in Supplementary Information 3.3, 5.3 and 6.3, and Supplementary Fig. 8–9.

**Software availability.** GSIM 6.0 models is available from website http://www.i4ide.org/content/wpapers.html. Information about IMS is available from website http://www.sfu.ca/emrg/Our_Research/policy-modelling.html. Information about TIMES is available from the website https://www.kanors-emr.org/.

**Reporting summary.** Further information on research design is available in the Nature Research Reporting Summary linked to this article.

## Data availability

The processed data for PV trade, embodied carbon flow, future energy mix projections, emission reduction potential and the generated results of trade barrier impacts simulation are provided in Source data file (Source Data.xlsx). The PV product trade data used in this study are available from multilateral institutional databases, including, the UN Comtrade database (https://comtrade.un.org/), and from databases of various countries/economies' customs and trade departments, e.g., USITC (https://dataweb.usitc.gov/). The life cycle inventory data used in this study are available from Ecoinvent database (https://www.ecoinvent.org/home.html). The data for power generation carbon emission factors calculations for various countries/economies are from

various official statistics, e.g., EU countries (https://www.eea.europa.eu/data-and-maps/data/co2-intensity-of-electricity-generation). Other scattered supporting data sources have been linked to the literature or websites cited in the paper. Source data are provided with this paper.

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

## Acknowledgements

The research presented in this paper benefitted from funding from the National Natural Science Foundation of China (grant no. 42071270) and the National Social Science Fund Key Project (grant no. 17AYJ011). This work was also funded by Global Energy Internet Group Co., Ltd. (grant no. SGTYHT/18-JS-206). In addition, Peng Song was supported by the Fundamental Research Funds for the Central Universities (grant no. 2019CDQYGG022). Eric Zusman was supported by funding from the Ministry of Environment, Japan, as part of its work on short-lived climate pollutants, and by Clean Air Asia as part of its work on the Integrated Better Air Quality Programme. We sincerely thank Professor Mark Jaccard of Simon Fraser University who provides IMS modelling support to the present research.

## Author contributions

X.M. and P.S. designed the study and guided the writing. M.W. led the data collection and undertook TFM analysis and the GSIM modelling and writing. Y.X. and K.T. undertook the IMS modelling. J.L. and Z.G. undertook the data collection and processing, GSIM and TIMES modelling. Z.L. was involved in GSIM model simulation and discussion. E.Z. contributed to the elaboration of the policy implications and literature review. M.W., X.M., Y.X., J.L., P.S., Z.L., Z.G., K.T. and E.Z. all contributed to the elaboration and writing of the article.

## Competing interests

The authors declare no competing interests.
