## [Peer Review File · Nature Communications]

Reviewer #1 (Remarks to the Author):

This paper tries to estimate the emissions reduction potential of PV of global PV product trade based on partial equilibrium analysis. In general, the topic is important. However, there are some major problems in your work.

Here are some specific comments:

1. The main tool of simulation analysis in this paper is GSIM model, which belongs to a short-term analysis. However, the author uses the simulation results of GSIM model to estimate the long-term environmental impacts of trade barriers of PV products. Is this reasonable? The author needs to explain this.
2. Two basic facts that the author should pay attention to when using GSIM model to simulate. First, in January, 2018, when the U.S. government announced that it would impose 201 tariff on imported PV products, it will set a duty-free quota of 2.5 GW, and impose 30% tariff on imports exceeding this quota in the first year, and the tax rates will decrease to 25%, 20% and 15% in the following three years; Second, on June 12, 2019, the Federal Trade Department of the United States ruled that three types of PV products, including double-sided photovoltaic modules assembled by double-sided batteries, 250-900w flexible glass fiber solar panels and some optical thin-film cell panels, were exempted from the 201 tariff from June 13, 2019.
3. This paper only evaluates the overall adverse impact of trade barriers on global carbon emissions, but ignores the differences in the impact of trade barriers on the carbon emission reduction of photovoltaic products exporting and importing countries. For example, although trade barriers are not conducive to the reduction of carbon emissions of countries such as the United States which impose high tariffs on photovoltaic products, they may be conducive to the reduction of carbon emissions of photovoltaic producing countries and exporting countries such as China. It is suggested that the author should evaluate the impact of trade barriers on carbon emissions of different countries, and then evaluate the overall impact from a global perspective.
4. Please give a more detailed introduction to the methods used in this paper to predict the carbon emission reduction potential of extra tariffs.

Reviewer #2 (Remarks to the Author):

Summary: This paper has one essential point, raising barriers to trade in solar photovoltaic products (PV) will lead to a reduction in overall penetration of solar energy and ceteris paribus an increase in CO₂ emissions. This is non-controversial and obvious since trade barriers necessarily raise the price of the related products and assuming that end-users are price sensitive would necessarily lead to a decline in demand. (Not mentioned by the authors is that trade barriers could also potentially slow down research and development and hence hinder improvements in the products themselves or their production.) The paper's headline number is that trade barriers could reduce the carbon emission mitigation potential of PV between 5-8 Gt CO₂e (to be related to a total carbon budget of between 500-1500 Gt CO₂e depending on the scenarios between 2020 and 2050). The paper highlights two other points: (1) one needs to take into account the full life cycle of products and thus PV has some CO₂ emissions linked to its use (for example in its production and transportation); and (2) that there are also other environmental impacts of PVs versus the alternatives such as emissions from particulate matter.

Comments:

The authors provide a wealth of facts and detailed analysis, and the headline number is of plausible magnitude. At the same time, the paper is 33 pages long and one suspects that a reasonable range of the impacts of a tariff on total demand could be had with a simple back of the envelope calculation. How much does the tariff (or tariffs) raise the average end-user price of the relevant goods? What is the elasticity of demand for the product?

The paper abstracts from many other relevant factors—particularly going out to 2050. (1) How likely are today's trade patterns to hold over the next 30 years? Would China continue to have a dominant position on the markets irrespective of the trade wars? Would its aging population and rapidly rising middle-class price China out of the market to the benefit of competing producers? (2) To what extent is the trade war a temporary phenomenon and a function of the current political environment? Should the world's nations get serious about decarbonizing the global economy, wouldn't they seek policies that accelerate PV penetration? (3) What are the substitute technologies and how substitutable are they with PV? (4) What other factors will influence the future of PV? For example, the aforementioned R&D and product innovation.

There is some degree of sensitivity analysis with respect to the scenarios, but it would be useful to provide a more thorough and systematic sensitivity analysis as regards the key parameters: future trade shares, supply, demand and substitution elasticities.

The paper is a very difficult read. It is chalk full of numbers and figures (some of which are mislabeled), using more significant digits than necessary, and rarely contextualized—are the numbers large or small? and relative to what?

Reviewer #3 (Remarks to the Author):

The manuscript entitled "Barriers on PV trade will impede global carbon mitigation and local pollutant emissions reduction" is focused on the emission potential reduction of global PV trade and the effect of extra trade barriers on PV product globally. The topic is very interesting and timely. However, crucial assumptions and data sources should be revised in the method section for the current and future scenarios under analysis. Below I list my specific comments/concerns.

In this study, PV life-cycle carbon emissions are obtained from various existing literature studies (some of them are not the latest available as detailed below in this report), reflecting the 2009-2013 silicon PV technology state of the art. In the developed future scenarios (from 2017-2050) the same PV carbon emissions are assumed. This assumption is a weak point because PV life-cycle carbon emissions are expected to decrease in the medium-long term mainly due to improvements in manufacturing processes. Specifically:

- (i) reduced silicon wafer thickness;
- (ii) reduced material usage (such as silicon, silver content in the metallization paste, copper), and kerf losses;
- (iii) electricity consumption reductions;
- (iv) PV module power conversion efficiency improvements.

As an example, please see the latest Fraunhofer report 2020, available at: <https://www.ise.fraunhofer.de/content/dam/ise/de/documents/publications/studies/Photovoltaics-Report.pdf>, which shows silicon usage reduction ranging from 150 g/Wp in 2013 to 80 g/Wp in 2019 as well as efficiency improvements in silicon-based PV module over the last years.

Please check also the following IEA PVPS Task 12 Report, in which are listed potential future changes (such as reductions in kerf loss, wafer thickness, carbon emissions and power conversion efficiency improvements):

Frischknecht, R., Itten, R., Wyss, F., Blanc, I., Heath, G.A., Raugei, M., Sinha, P. and Wade, A., 2015. Life cycle assessment of future photovoltaic electricity production from residential-scale systems operated in Europe (No. NREL/TP-6A20-73849). Available at: <https://iea-pvps.org/key-topics/iea-pvps-task-12-life-cycle-assessment-of-future-photovoltaic-electricity-production-from-residential-scale-systems-operated-in-europe-2015/>

In this study, these potential changes are not discussed or included, such as providing a sensitivity analysis (or uncertainty analysis) for potential future emission changes due to production improvements. Also, it is not captured/discussed potential changes in the electricity grid mix compositions (such as increased penetration of renewable energies), which influence the amount

of primary energy required for each PV production process as well as the associated carbon emissions.

Page 2 Lines 31-35

It is stated that "studies have been conducted varying from calculating the lifecycle emissions and emission reduction potential of PV products (8-21) to PV product trade patterns and structures (22) to the emissions embodied in the bilateral PV product trade²³ and to the environmental impacts of the PV product trade (13,14,24). Studies have also been carried out in various countries (15-21,24), concluded that PV power generation could indeed help to reduce carbon and local air pollutant emissions obviously."

The authors cite here previous PV life cycle studies, 7 of which are focused on China, 1 in the UK, 1 in Turkey, and 1 in the United States, and most of them are based on crystalline silicon technology. Since the scope of this paper is "global" and it includes future scenarios, it would be appropriate to extend the existing background references, integrating them with other studies that are focused also on:

- (i) PV produced and/or installed in other geographical areas,
- (ii) comparison between PV life cycle carbon emissions with other electricity generation technologies (fossil fuels and renewables) in specific electricity grid mixes,
- (iii) complete PV systems, including balance of system (BOS) (structures, inverters, cables, transformers).
- (iv) the additional life cycle contribution of potential storage and/or batteries needed for future scenarios with large renewable electricity penetration.
- (v) PV end-of-life life-cycle contribution.

Page 2 Lines 45-47

"previous studies concentrated on only a specific PV product and a limited subset of countries/regions without full consideration for the whole production and supply chain from a global perspective"

This statement is not correct. Please check the following literature studies:

- 1) <https://doi.org/10.1016/j.energy.2018.08.051>
- 2) <https://doi.org/10.1016/j.enpol.2017.11.062>
- 3) <https://doi.org/10.3390/en13153934>
- 4) <https://doi.org/10.1002/ente.201901146>
- 5) <https://doi.org/10.3390/en9080622>
- 6) <https://doi.org/10.1016/j.solmat.2013.08.037>
- 7) <https://doi.org/10.1002/pip.3316>
- 8) <https://doi.org/10.1016/j.ecolind.2016.03.028>
- 9) <https://doi.org/10.1002/pip.2650>
- 10) <https://doi.org/10.1016/j.solener.2009.10.002>

Lines 78-88

Please list the references for your trade flow matrix (TFM).

Page 20 Lines 481-482

It is stated that "PV power generation is relatively clean and near zero-emissions in its application, but certain amounts of GHGs and pollutants are emitted during the PV products production process"

Life cycle analysis takes into the account not only the production processes, but the product's full life cycle from the extraction of resources and the production of raw materials, to manufacturing, distribution, use and re-use, maintenance, and finally recycling and disposal of the final product – including transportation and use of energy carriers.

Please check the following references, which are focused on LCA guidelines and standardization as well as specific PV LCA guidelines:

- 1) J. B. Guinée, *Int. J. Life Cycle Assess.*, 2002, 7, 311-313F.
- 2) Consoli, D. Allen, I. Boustead, N. de Oude, J. Fava, R. Franklin, A. A. Jensen, R. Parrish, R.

Perriman, D. Postlethwaite, B. Quay, J. Séguin, B. Vigon, In 1993 Proceeding of the SETAC Workshop, Sesimbra, Portugal.

3) International Organization for Standardization, Standard ISO, 14040 Environmental Management – Life Cycle Assessment – Principles and Framework, 2006, <https://www.iso.org/standard/37456.html>

4) International Organization for Standardization, Standard ISO, 14044 Environmental Management – Life Cycle Assessment – Requirements and guidelines, <https://www.iso.org/standard/38498.html>

5) V. M. Fthenakis, R. Frischknecht, M. Raugei, H. C. Kim, E. Alsema, M. Held and M. de Wild-Scholten, Methodology Guidelines on Life Cycle Assessment of Photovoltaic Electricity, 2nd edition, Int Energy Agency PVPS Task 12, 2011, Report IEA-PVPS T12-03:2011.

6) R. Frischknecht, G. Heath, M. Raugei, P. Sinha, M. de Wild-Scholten, V. M. Fthenakis, H. C. Kim, E. Alsema, M. Held, Methodology Guidelines on Life Cycle Assessment of Photovoltaic Electricity, 3rd edition, Int. Energy Agency PVPS Task 12, 2016, Report IEA-PVPS T12-06:2016.

Page 20, lines 481-493.

Are the balance of system (BOS) emission factors included in the analysis (inverters, transformers, cables, structures)? It is a crucial part of any PV energy system, and this contribution should be taken into the account for PV power generation life-cycle emissions, according to the methodology guidelines of PV LCA.

Excluding the BOS emissions is a weak point.

Please check the 2011 and 2016 PV LCA guidelines:

- V. M. Fthenakis, R. Frischknecht, M. Raugei, H. C. Kim, E. Alsema, M. Held and M. de Wild-Scholten, Methodology Guidelines on Life Cycle Assessment of Photovoltaic Electricity, 2nd edition, Int Energy Agency PVPS Task 12, 2011, Report IEA-PVPS T12-03:2011.

- R. Frischknecht, G. Heath, M. Raugei, P. Sinha, M. de Wild-Scholten, V. M. Fthenakis, H. C. Kim, E. Alsema, M. Held, Methodology Guidelines on Life Cycle Assessment of Photovoltaic Electricity, 3rd edition, Int. Energy Agency PVPS Task 12, 2016, Report IEA-PVPS T12-06:2016.

Page 20, Lines 487-494

The authors state that they obtained the PV emission factors from other life cycle studies. Specifically, it is stated that they are based on:

- Hou et al., 2016 for China (assuming the same emission factor for Hong Kong Taiwan, Russia, India and Turkey) [9];
- Wetzel et al., 2015 for Germany (assuming the same emission factor for Europe) [72];
- Ecoinvent database version 3 for the USA [73];
- Kim et al., 2014 for Korea (assuming the same for Japan and Oceania) [74];
- Luo et al., 2018 for Sinagapore (assuming the same for all ASEAN countries) [75];

In the supplementary information section 3 the considered emission factors taken from the literatures are listed, and it is stated that “because regional LCA studies of PV products were quite limited, we adopted the latest and up-to-date LCA emission coefficients available to carry out embodied emission calculation”.

However, Ecoinvent database version 3 does not provide directly the PV “emission factors”, but Ecoinvent is a database with specific processes. Emission factors should be calculated, considering specific processes (eventually updating them based on the geographical locations and their current grid mixes), and then applying an impact assessment method (such as CML). What processes from Ecoinvent were considered in this analysis? What are the life cycle inventories used in the considered Ecoinvent processes?

The associated life cycle inventories are probably outdated, and they are not the latest.

Please check this 2016 manuscript in which emission factors are calculated for Europe, USA and China, considering their respective electricity grid mix compositions:

<https://doi.org/10.3390/en9080622>.

Table 2 in the supplementary information shows the assumed carbon emissions for 4 products that are silicon, silicon wafer, solar cell, and PV module. However, there is a lack of transparency for reproducing the numbers. My main questions are:

1) What processes/contributions are included in each stage? Are the aluminum frame and encapsulations included in the PV module contribution?

It would be useful to add a flow diagram to be transparent on what is included/excluded.

2) What is the considered silicon type (single-crystalline or multi-crystalline)? They have different efficiencies, energy demand and carbon emissions. Specifically, the Cz process in the single-silicon cell is more energy demanding and more impactful in terms of carbon emissions (<https://doi.org/10.3390/en9080622>).

3) The USA wafer carbon emissions listed are 302.83 kgCO₂/kg wafer, while wafers produced in China cause 150.29 kg CO₂/kg wafer. Why the USA emission factors is approximately twice as that for China?

On the contrary, the carbon emissions of US silicon and US solar cells are lower compared to the ones in China (79.34 kg CO₂/kg silicon compared to 129.86 kg CO₂/kg silicon and 0.79 kgCO₂eq/Wcell compared to 0.96 kgCO₂eq/ Wcell)? What is the reason for such differences?

5) Solar cell and PV module carbon emissions are expressed per W cell. What are the assumed efficiencies? (Please see the Fraunhofer report in which the latest commercial c-Si efficiencies are listed on pag. 29:

<https://www.ise.fraunhofer.de/content/dam/ise/de/documents/publications/studies/Photovoltaics-Report.pdf>)

6) The function units in Table 2 are different (kg and W). How did you combine those? What is the material usage (kg per silicon used per wafer and kg of wafer used per W cell)?

7) What is the thickness of the wafers?

8) The BOS contribution is not listed. BOS emissions factor ranges from 251-280 kgCO₂eq/kWp (considering power conversion efficiencies of 20%-18%) (see figure 4 in the following reference: <https://www.mdpi.com/1996-1073/9/8/622>)

Page 21 lines 499-507

Air pollutants (PM₁₀, NO_x and SO₂) of PV trade were assumed from the literature.

For China ref. n. 12 is assumed as data source, while for all the other countries Ecoinvent database is used. To be consistent it would be better to assume the same source of data.

Air pollutants are listed in Table 3 and 4 in the supplementary information.

Please clarify which countries are included in "other".

Also, other questions are?

- Why do Europe and America have the same local pollutant emissions? The respective electricity grid mixes are different.

- What are the efficiencies assumed per W?

- Wafer thickness?

- In Table 4 (emissions of PV in China) two years are assumed as reference (2014 and 2017), while in Table 3 there is only one year. What is the reason?

- Please explain the reason for huge differences between 2014 and 2017 values in Table 4 (such as PM₁₀ from solar module: 0.103 g/W cell for 2014 compared to 0.024 g/W cell for 2017).

- It is also not clear the reason for differences between values in table 3 compared to those in table 4. As an example, NO_x associated to solar module in Europe are 0.721 kg/W module (which is similar to the value for "other" countries), while in China in 2014 are 1.389 and in 2017 are 0.215.

Page 21, lines 494-495

It is stated that "the emissions coefficients were converted into emissions per US\$". Please provide further details on the conversion factors that are used with references and explanation.

Page 21 lines 525-526

Please provide further details on the solar power generation potential predicted for the installed capacity increment.

Page 22 line 532

In your second scenario, did you replace all the thermal power generation by PV?

Figure 2 a page 8:

the pie chart shows the CO₂ percentage for silicon, wafer, and C&M? Please provide explanation on C&M contribution? Does it stand for cell and module?

Figures 2c and 2d: the legend is difficult to read.

Figure 3:

Please provide explanation for local pollutant reduction from 2014 to 2017.

Page 10, Lines 226-229:

Please detail the assumptions on thermal power generation replacement.

Page 11 lines 265-266

It is stated that "solar power replacement of fossil fuel-fired generation also leads to local air pollutant emissions abatement".

As stated above in this report, the contribution of balance of systems should be added because it is a crucial part of PV systems for producing electricity.

Also, electricity grids with a large penetration of PV (such as in your future scenarios) need storage systems, and their life-cycle emissions should be considered.

Please see this 2020 manuscript: <https://doi.org/10.1002/ente.201901146> and the other PV LCA literature studies on storage listed above.

Other comments:

- In the manuscript, it is used "lifecycle" as a one word, but it is more common "life cycle" as two words in the LCA literature.

- Often, it is implicitly assumed in the manuscript that silicon is the only available technologies for PV panels, and this is incorrect. Some examples are:

Lines 35-38

"Although PV power generation has nearly 'zero emissions' in its operation, pollutant emissions cannot be ignored when considering the whole lifecycle of PV products--from ore mining, silicon and silicon wafer processing, and solar cell and PV panel producing"

This statement implicitly assume that PV systems can be only be fabricated with silicon and it is incorrect. It should be specified that your statement refers to PV crystalline-based technologies. Although the main commercial PV technology is currently based on silicon (~94% of the market) – there are also other PV technologies commercially available, such as thin film cadmium telluride (CdTe) and copper indium gallium (di)selenide/(di) sulfide (CIGS) technologies [1]. Thin film technologies are more recent PV technologies compared to silicon-based architectures, and they offer potential benefits compared to wafer technology, including ~100x thinner absorber layers and opportunities for low cost. Thin film technologies also show reductions in life cycle carbon emissions and other environmental impacts [2]. There are also PV emerging technologies – such as perovskite PV (single junction and tandem architectures) – that have been attracting intensive attention due to notable laboratory efficiency improvements, and that may be implemented to industrial scale [3]. Their life-cycle impacts have been also evaluated in recent studies [4, 5, 6, 7, 8].

[1] Solar Power Europe. Global market outlook for solar power 2015-2019; solar power Europe. Brussels, Belgium: European Photovoltaic Industry Association; 2015. p. 2015.

[2] Fthenakis, V., Athias, C., Blumenthal, A., Kulur, A., Magliozzo, J. and Ng, D., 2020. Sustainability evaluation of CdTe PV: An update. *Renewable and Sustainable Energy Reviews*, 123, p.109776.

[3] Li, Z., Klein, T.R., Kim, D.H., Yang, M., Berry, J.J., van Hest, M.F. and Zhu, K., 2018. Scalable

fabrication of perovskite solar cells. *Nature Reviews Materials*, 3(4), pp.1-20.

[4] Gong J, Darling SB, You F. Perovskite photovoltaics: Life-cycle assessment of energy and environmental impacts. *Energy Environ Sci* 2015. doi:10.1039/c5ee00615e.

[5] Celik I, Phillips AB, Song Z, Yan Y, Ellingson RJ, Heben MJ, et al. Environmental analysis of perovskites and other relevant solar cell technologies in a tandem configuration. *Energy Environ Sci* 2017;10:1874–84. doi:10.1039/c7ee01650f.

[6] Billen, P., Leccisi, E., Dastidar, S., Li, S., Lobaton, L., Spatari, S., Fafarman, A.T., Fthenakis, V.M. and Baxter, J.B., 2019. Comparative evaluation of lead emissions and toxicity potential in the life cycle of lead halide perovskite photovoltaics. *Energy*, 166, pp.1089-1096.

[7] Leccisi, E. and Fthenakis, V., 2019. Critical review of perovskite photovoltaic life cycle environmental impact studies. In 2019 IEEE 46th Photovoltaic Specialists Conference (PVSC) (Vol. 2, pp. 1-6).

[8] Celik, I., Song, Z., Cimaroli, A.J., Yan, Y., Heben, M.J. and Apul, D., 2016. Life Cycle Assessment (LCA) of perovskite PV cells projected from lab to fab. *Solar Energy Materials and Solar Cells*, 156, pp.157-169.

Lines 428-429

Please specify that it is referred only to crystalline-based technologies.

Lines 468-470 and lines 136-137

This is a repetition. The following sentence "this study converted all the currencies to US dollars first and then converted them to 2010 US dollars for comparability" is repeated two times.

Title: Barriers on PV trade will impede global carbon mitigation

Response to the reviewers' comments

Reviewer #1:

This paper tries to estimate the emissions reduction potential of PV of global PV product trade based on partial equilibrium analysis. In general, the topic is important. However, there are some major problems in your work.

Here are some specific comments:

Reviewer comments:

1. The main tool of simulation analysis in this paper is GSIM model, which belongs to a short-term analysis. However, the author uses the simulation results of GSIM model to estimate the long-term environmental impacts of trade barriers of PV products. Is this reasonable? The author needs to explain this.

Response to reviewer comments:

Many thanks for the comments and suggestion. We have revised the manuscript and make up for this defect by deploying a dynamic model (integrated modelling system, IMS) in combination with GSIM model. GSIM model is used to simulate trade barriers impacts on global PV trade pattern, while IMS model is applied to project the long-term global and country/economy specific power production and supply composition and the relevant environmental impacts up to 2060. Although dynamic CGE model such as GTAP can also project long-term impacts of tariff rate imposition, it is very hard to separate and simulate the PV goods trade, which are only a small fraction of several sectors, such as Mineral products, Computer, electronic and optical products, and Electrical products. And it is not convenient for CGE to describe and simulate the detailed technology competition in the power generation sector. The detailed content is shown in Results “Projection to power market share and carbon emissions reduction potential of PV application” and “Barriers to the PV product trade would impede global emissions reduction potential”, and in the Methods and Supplementary Information 5 and 6.

Reviewer comments:

2. Two basic facts that the author should pay attention to when using GSIM model to simulate. First, in January, 2018, when the U.S. government announced that it would impose 201 tariff on imported PV products, it will set a duty-free quota of 2.5 GW, and impose 30% tariff on imports exceeding this quota in the first year, and the tax rates will decrease to 25%, 20% and 15% in the following three years; Second, on June 12, 2019, the Federal Trade Department of the United States ruled that three types of PV products, including double-sided photovoltaic modules assembled by double-sided batteries, 250-900w flexible glass fiber solar panels and some optical thin-film cell panels, were exempted from the 201 tariff from June 13, 2019.

Response to reviewer comments:

Many thanks for the comments and suggestion. In revising the GSIM model simulation and the policy scenario setting, we have tried our best to take the real world tariff barrier including the USA PV goods import tariff policy that the reviewer mentioned into consideration. However this study is to disclose the general rule that, trade barrier on PV goods in the long-run will inevitably impede global carbon reduction potential, rather than to evaluate the USA PV goods trade policy in very detail in a short term. Thus we set three policy scenarios based on the ongoing, foreseeable and possible tariff imposition situations, and also took uncertainties of tariff rates adjustment into account (Supplementary Information 6). We would think that the very detailed PV goods tariff rate changes, for example, the exemption of thin-film cell panels, are beyond the research scope of this study, but we are willing to do further analysis in some other researches in the near future.

Reviewer comments:

3. This paper only evaluates the overall adverse impact of trade barriers on global carbon emissions, but ignores the differences in the impact of trade barriers on the carbon emission reduction of photovoltaic products exporting and importing countries. For example, although trade barriers are not conducive to the reduction of carbon emissions of countries such as the United States which impose high tariffs on photovoltaic products, they may be conducive to the reduction of carbon emissions of photovoltaic producing countries and exporting countries such as China. It is suggested that the author should evaluate the impact of trade barriers on carbon emissions of different countries, and then evaluate the overall impact from a global perspective.

Response to reviewer comments:

Many thanks for the great comments and suggestion. In the revision, we have evaluated the impacts of trade barriers on carbon emissions of different countries, and then evaluate the overall impacts from a global perspective in Results section “Barriers to the PV product trade would impede global emissions reduction potential”, for example,

“ In escalated tariff scenarios TBS1 and TBS2, countries/economies that exert higher PV goods import tariff rates will experience larger carbon reduction potential losses; e.g., India and the USA are expected to lose net carbon reduction potential by 5.25-5.44 and 2.17-2.48 GtCO₂e (SST), respectively. Although trade barriers are not conducive to the reduction of carbon emissions of countries such as India and the United States which impose high tariffs on PV products import, they are conducive to the reduction of carbon emissions of PV producing and exporting countries such as the Republic of Korea and China, whose net carbon reduction potential will increase by 0.03-0.05 and 0.42-0.58 GtCO₂e (SST), respectively. However, the overall impacts of trade barrier on PV goods cause the global carbon emission reduction potential to decrease.

Escalated tariff scenarios TBS1 and TBS2 will decrease the global net carbon reduction potential by 3.71-6.34% or 3.26-6.77 GtCO₂e (SSG-SST) and 4.37-6.26% or 3.22-7.98 GtCO₂e (SSG-SST), respectively. (see Fig. 7b)”

Reviewer comments:

4. Please give a more detailed introduction to the methods used in this paper to predict the carbon emission reduction potential of extra tariffs.

Response to reviewer comments:

Many thanks for the comments and suggestion. In the revision, we have elaborated the methods in more details to describe how to predict/simulate tariff impacts on carbon emission reduction in Methods section “A technology-based integrated dynamic model to project power market share” and “Tariff barrier scenarios and computable partial equilibrium model simulation”, and also in Supplementary Information 5 and 6.

Reviewer #2:

Reviewer comments:

The authors provide a wealth of facts and detailed analysis, and the headline number is of plausible magnitude. At the same time, the paper is 33 pages long and one suspects that a reasonable range of the impacts of a tariff on total demand could be had with a simple back of the envelope calculation. How much does the tariff (or tariffs) raise the average end-user price of the relevant goods? What is the elasticity of demand for the product?

Response to reviewer comments:

Many thanks for reminding us to provide the information of the impacts of escalated tariffs on the average price of the PV goods, which is the key to affect demand and supply and applications of PV solar cells and modules. Based on GSIM model simulation, PV products prices changes were obtained and are listed in Supplementary Information 6.2. The demand elasticity parameters involved in the GSIM model are also given in Supplementary Information 6.1 Supplementary Table 8.

Reviewer comments:

The paper abstracts from many other relevant factors—particularly going out to 2050. (1) How likely are today’s trade patterns to hold over the next 30 years? Would China continue to have a dominant position on the markets irrespective of the trade wars? Would its aging population and rapidly rising middle-class price China out of the market to the benefit of competing producers? (2) To what extent is the trade war a temporary phenomenon and a function of the current political environment? Should the world’s nations get serious about decarbonizing the global economy, wouldn’t they seek policies that accelerate PV penetration? (3) What are the substitute technologies and how substitutable are they with PV? (4) What other factors will influence the future of PV? For example, the aforementioned R&D and product innovation.

Response to reviewer comments:

Many thanks for reminding us that this paper abstracts from many relevant factors.

- (1) Trade pattern is indeed an important factor relevant to the projection results. We considered that trade pattern is largely decided by the differences in factor endowment of various countries/economies which are relatively stable, e.g., though East Asia countries including China have the problem of aging population and rapidly rising middle-class, their comparative advantages in PV goods supply over other regions should last long, and thus we would assume a stable trade pattern to facilitate the trade barrier scenario analysis. For the power demand predictions for various countries/economies in IMS model simulations, the population and

income factors have been considered in the parameter setting.

- (2) In this study, we would like to disclose the fact/rule that, ongoing, foreseeable and possible trade barriers can impact on global PV trade and harm the global carbon mitigation capacity, and try to call on for freer trade of PV goods. We and many scholars are very much concern that trade war is not simply a temporary phenomenon, but a function of the current and long-lasting political environment in which the USA-Sino competitions are intensified, which is not conducive to global joint effort fighting against climate change. Recently, major carbon emission countries/economies, including China, the USA, Japan, EU, among others, made serious promise to achieve carbon neutrality by 2050 or 2060. We would like to take these promises as serious ones and have taken these factors into our dynamic model (IMS) simulation, referring to national energy plans, energy development outlooks, NDC target commitments, long-term energy strategic plans and decarbonisation pathway research reports of various countries/economies.
- (3) and (4) This study adopted IMS model to simulate the dynamic evolution of power generation and supply market in various countries/economies from 2017 to 2060, with an emphasis on the competition of solar PV with other power generation technologies, including coal, gas, oil, nuclear, solar thermal, geothermal, marine, hydropower, onshore wind, offshore wind, biofuel, biogas and waste incineration. The technology progress brought about by R&D were reflected in the production carbon emission coefficient decreases, PV panel conversion efficiency improvement, and performance rate increases which in turn will reduce the embodied carbon and increase the net carbon emission reduction potential of globally traded PV products (Supplement Information 3.3).

Reviewer comments:

There is some degree of sensitivity analysis with respect to the scenarios, but it would be useful to provide a more thorough and systematic sensitivity analysis as regards the key parameters: future trade shares, supply, demand and substitution elasticities.

Response to reviewer comments:

Many thanks for the comments and suggestion. In the revision, we have added more detailed analysis to test the sensitivities of simulation results, such as, future trade (imports and exports) volume and structure (shares) and consumptions (demand and supply), with respect to the tariff rates changes under different scenarios, varied substitution elasticities settings, among others (Supplementary Information 6).

Reviewer comments:

The paper is a very difficult read. It is chalk full of numbers and figures (some of which are mislabeled), using more significant digits than necessary, and rarely contextualized—are the numbers large or small? and relative to what?

Response to reviewer comments:

Many thanks for the comments and suggestion. In the revision, we have improved the manuscript by deleting redundant numbers, unnecessary tables and figures, polishing the language, highlighting

the most important results, and improving the contexture. We hope the revised manuscript is easier and better to read.

Reviewer #3:

The manuscript entitled “Barriers on PV trade will impede global carbon mitigation and local pollutant emissions reduction” is focused on the emission potential reduction of global PV trade and the effect of extra trade barriers on PV product globally.

The topic is very interesting and timely. However, crucial assumptions and data sources should be revised in the method section for the current and future scenarios under analysis.

Below I list my specific comments/concerns.

Reviewer comments:

In this study, PV life-cycle carbon emissions are obtained from various existing literature studies (some of them are not the latest available as detailed below in this report), reflecting the 2009-2013 silicon PV technology state of the art. In the developed future scenarios (from 2017-2050) the same PV carbon emissions are assumed. This assumption is a weak point because PV life-cycle carbon emissions are expected to decrease in the medium-long term mainly due to improvements in manufacturing processes. Specifically:

- (i) reduced silicon wafer thickness;
- (ii) reduced material usage (such as silicon, silver content in the metallization paste, copper), and kerf losses;
- (iii) electricity consumption reductions;
- (iv) PV module power conversion efficiency improvements.

As an example, please see the latest Fraunhofer report 2020, available at: <https://www.ise.fraunhofer.de/content/dam/ise/de/documents/publications/studies/Photovoltaics-Report.pdf>, which shows silicon usage reduction ranging from 150 g/Wp in 2013 to 80 g/Wp in 2019 as well as efficiency improvements in silicon-based PV module over the last years.

Please check also the following IEA PVPS Task 12 Report, in which are listed potential future changes (such as reductions in kerf loss, wafer thickness, carbon emissions and power conversion efficiency improvements):

Frisknecht, R., Itten, R., Wyss, F., Blanc, I., Heath, G.A., Raugei, M., Sinha, P. and Wade, A., 2015. Life cycle assessment of future photovoltaic electricity production from residential-scale systems operated in Europe (No. NREL/TP-6A20-73849). Available at: <https://iea-pvps.org/key-topics/iea-pvps-task-12-life-cycle-assessment-of-future-photovoltaic-electricity-production-from-residential-scale-systems-operated-in-europe-2015/>

In this study, these potential changes are not discussed or included, such as providing a sensitivity analysis (or uncertainty analysis) for potential future emission changes due to production improvements. Also, it is not captured/discussed potential changes in the electricity grid mix compositions (such as increased penetration of renewable energies), which influence the amount of primary energy required for each PV production process as well as the associated carbon emissions.

Response to reviewer comments:

Many thanks for the comments and suggestion. In the revision, we have fully considered the suggested aspects, i.e., reduced silicon wafer thickness; reduced material usage (such as silicon, silver content in the metallization paste, copper) and kerf losses; electricity consumption reductions; potential changes in the electricity grid mix, etc., taken the technology progress of PV products manufacture into consideration to update the emission coefficients in the Methods part, and added an uncertainty analysis in Supplementary Information 3.3 to address the sensitivity of embodied carbon emission and the net carbon emission reduction potential of global traded PV products with respect to the carbon emission coefficients (CE) decreases, PV module power conversion efficiency (CE) improvements, and PV performance rate (PR) increases. In Supplementary Information 5 we also addressed the projected electricity power supply composition in various countries/economies and the world, and its impacts on the PV production process carbon emission were reflected in the carbon emission coefficients (CE) decreases.

Reviewer comments:

It is stated that “studies have been conducted varying from calculating the lifecycle emissions and emission reduction potential of PV products (8-21) to PV product trade patterns and structures (22) to the emissions embodied in the bilateral PV product trade (23) and to the environmental impacts of the PV product trade (13,14,24). Studies have also been carried out in various countries (15-21, 24), concluded that PV power generation could indeed help to reduce carbon and local air pollutant emissions obviously.”

The authors cite here previous PV life cycle studies, 7 of which are focused on China, 1 in the UK, 1 in Turkey, and 1 in the United States, and most of them are based on crystalline silicon technology. Since the scope of this paper is “global” and it includes future scenarios, it would be appropriate to extend the existing background references, integrating them with other studies that are focused also on:

- (i) P V produced and/or installed in other geographical areas,
- (ii) comparison between PV life cycle carbon emissions with other electricity generation technologies (fossil fuels and renewables) in specific electricity grid mixes,
- (iii) complete PV systems, including balance of system (BOS) (structures, inverters, cables, transformers).
- (iv) the additional life cycle contribution of potential storage and/or batteries needed for future scenarios with large renewable electricity penetration.
- (v) PV end-of-life life-cycle contribution.

Response to reviewer comments:

Many thanks for the comments and suggestion. In revision, we have extended the existing background literature to studies on PV produced and/or installed in other geographical areas, PV life cycle carbon emissions, complete PV systems (including balance of system (BOS), storage and/or batteries, the additional life cycle contribution of potential storage and/or batteries, PV end-of-life life-cycle contribution, etc., in the Introduction section, for example, “Studies have been conducted ranging from calculating the life cycle emissions and emissions reduction potential of PV products (8,10-16,19-24) to PV systems including balance of system (BOS) and storage and/or batteries (25-29) and their end-of-life emission contribution (30,31), life cycle impact comparisons between solar power and other power generation technologies (32,33), and the impacts of PV

application on future electricity grids (34,35) in various geographical areas.” We have also used the updated literature to calibrate the emission coefficients and to calculate the carbon emission and carbon reduction potential.

Reviewer comments:

Page 2 Lines 45-47

“previous studies concentrated on only a specific PV product and a limited subset of countries/regions without full consideration for the whole production and supply chain from a global perspective”

This statement is not correct. Please check the following literature studies:

- 1) <https://doi.org/10.1016/j.energy.2018.08.051>
- 2) <https://doi.org/10.1016/j.enpol.2017.11.062>
- 3) <https://doi.org/10.3390/en13153934>
- 4) <https://doi.org/10.1002/ente.201901146>
- 5) <https://doi.org/10.3390/en9080622>
- 6) <https://doi.org/10.1016/j.solmat.2013.08.037>
- 7) <https://doi.org/10.1002/pip.3316>
- 8) <https://doi.org/10.1016/j.ecolind.2016.03.028>
- 9) <https://doi.org/10.1002/pip.2650>
- 10) <https://doi.org/10.1016/j.solener.2009.10.002>

Response to reviewer comments:

Many thanks for the comments and suggestions. In the revision, we have deleted the inappropriate statement and cited most of the literatures recommended by the reviewer.

Reviewer comments:

Lines 78-88

Please list the references for your trade flow matrix (TFM).

Response to reviewer comments:

Many thanks for the suggestion. We have listed the detailed literatures and data sources for building the trade flow matrix in Method section titled “**Multilateral PV product trade and trade flow matrix (TFM) construction**”.

Reviewer comments:

Page 20 Lines 481-482

It is stated that “PV power generation is relatively clean and near zero-emissions in its application, but certain amounts of GHGs and pollutants are emitted during the PV products production process” Life cycle analysis takes into the account not only the production processes, but the product’s full life cycle from the extraction of resources and the production of raw materials, to manufacturing, distribution, use and re-use, maintenance, and finally recycling and disposal of the final product – including transportation and use of energy carriers.

Please check the following references, which are focused on LCA guidelines and standardization as well as specific PV LCA guidelines:

- 1) J. B. Guinée, Int. J. Life Cycle Assess., 2002, 7, 311-313F.

2) Consoli, D. Allen, I. Boustead, N. de Oude, J. Fava, R. Franklin, A. A. Jensen, R. Parrish, R. Perriman, D. Postlethwaite, B. Quay, J. Séguin, B. Vigon, In 1993 Proceeding of the SETAC Workshop, Sesimbra, Portugal.

3) International Organization for Standardization, Standard ISO, 14040 Environmental Management – Life Cycle Assessment – Principles and Framework, 2006, <https://www.iso.org/standard/37456.html>

4) International Organization for Standardization, Standard ISO, 14044 Environmental Management – Life Cycle Assessment – Requirements and guidelines, <https://www.iso.org/standard/38498.html>

5) V. M. Fthenakis, R. Frischknecht, M. Raugei, H. C. Kim, E. Alsema, M. Held and M. de Wild-Scholten, Methodology Guidelines on Life Cycle Assessment of Photovoltaic Electricity, 2nd edition, Int Energy Agency PVPS Task 12, 2011, Report IEA-PVPS T12-03:2011.

6) R. Frischknecht, G. Heath, M. Raugei, P. Sinha, M. de Wild-Scholten, V. M. Fthenakis, H. C. Kim, E. Alsema, M. Held, Methodology Guidelines on Life Cycle Assessment of Photovoltaic Electricity, 3rd edition, Int. Energy Agency PVPS Task 12, 2016, Report IEA-PVPS T12-06:2016. Are the balance of system (BOS) emission factors included in the analysis (inverters, transformers, cables, structures)? It is a crucial part of any PV energy system, and this contribution should be taken into the account for PV power generation life-cycle emissions, according to the methodology guidelines of PV LCA.

Excluding the BOS emissions is a weak point.

Please check the 2011 and 2016 PV LCA guidelines:

- V. M. Fthenakis, R. Frischknecht, M. Raugei, H. C. Kim, E. Alsema, M. Held and M. de Wild-Scholten, Methodology Guidelines on Life Cycle Assessment of Photovoltaic Electricity, 2nd edition, Int Energy Agency PVPS Task 12, 2011, Report IEA-PVPS T12-03:2011.

- R. Frischknecht, G. Heath, M. Raugei, P. Sinha, M. de Wild-Scholten, V. M. Fthenakis, H. C. Kim, E. Alsema, M. Held, Methodology Guidelines on Life Cycle Assessment of Photovoltaic Electricity, 3rd edition, Int. Energy Agency PVPS Task 12, 2016, Report IEA-PVPS T12-06:2016.

Response to reviewer comments:

Many thanks for the comments and suggestions. We have carefully checked all the literatures recommended by the reviewer, and cited the most up-to-date PV LCA guidelines and revised the statement as “Although PV power generation is nearly ‘zero emissions’ during operation and could indeed help to substantially reduce carbon emissions (8-13), its emissions should not be ignored when the whole life cycle of PV products is considered (14-18).”

We have also added consideration of BOS and storage system related emissions in the revision, referring to the guidelines recommended by the reviewer and up-to-date researches to obtain emission coefficients and calculated the carbon emissions. The detailed information is shown in Supplementary Information 3.2 “**carbon embodied in BOS and storage system**”.

Reviewer comments:

Page 20, Lines 487-494

The authors state that they obtained the PV emission factors from other life cycle studies. Specifically, it is stated that they are based on:

- Hou et al., 2016 for China (assuming the same emission factor for Hong Kong Taiwan, Russia, India and Turkey) [9];

- Wetzel et al., 2015 for Germany (assuming the same emission factor for Europe) [72];
- Ecoinvent database version 3 for the USA [73];
- Kim et al., 2014 for Korea (assuming the same for Japan and Oceania) [74];
- Luo et al., 2018 for Sinagapore (assuming the same for all ASEAN countries) [75];

In the supplementary information section 3 the considered emission factors taken from the literatures are listed, and it is stated that “because regional LCA studies of PV products were quite limited, we adopted the latest and up-to-date LCA emission coefficients available to carry out embodied emission calculation”.

However, Ecoinvent database version 3 does not provide directly the PV “emission factors”, but Ecoinvent is a database with specific processes. Emission factors should be calculated, considering specific processes (eventually updating them based on the geographical locations and their current grid mixes), and then applying an impact assessment method (such as CML). What processes from Ecoinvent were considered in this analysis? What are the life cycle inventories used in the considered Ecoinvent processes?

The associated life cycle inventories are probably outdated, and they are not the latest.

Please check this 2016 manuscript in which emission factors are calculated for Europe, USA and China, considering their respective electricity grid mix compositions: <https://doi.org/10.3390/en9080622>.

Response to reviewer comments:

Many thanks for the kind comments and suggestions.

We used Ecoinvent database to inform emission coefficient of a PV product. When typing in a specific PV product name, e.g., silicon wafer, in the database interface, a life cycle inventory (LCI) considering emissions of all the upstream activities/processes of the product were presented, which are used as PV product emission coefficients to calculate the embodied carbon emissions. However, Ecoinvent only distinguishes the data applicability for Europe and the world and claims a long applicable period (for example 1.1.2005-12.31.2020).

Since as the reviewer indicated, the life cycle inventories in the Ecoinvent are outdated, in the revision, we adjusted the original coefficients from Ecoinvent according to the PV production technology improvement (e.g., reduced silicon wafer thickness; reduced material usage and kerf losses; electricity consumption reductions; potential changes in the electricity grid mix, etc.) described by the IEA PVPS trend report. The detailed information can be seen in Methods section “**Accounting embodied carbon emissions in PV products and their global trade**” and Supplement Information 3 “**Embodied carbon emission accounting**”.

Reviewer comments:

Table 2 in the supplementary information shows the assumed carbon emissions for 4 products that are silicon, silicon wafer, solar cell, and PV module. However, there is a lack of transparency for reproducing the numbers. My main questions are:

Response to reviewer comments:

Many thanks for the great comments. Please see in below the point-by-point response to the question:

- 1) What processes/contributions are included in each stage? Are the aluminum frame and

encapsulations included in the PV module contribution? It would be useful to add a flow diagram to be transparent on what is included/excluded.

Response to reviewer comments:

In the Supplementary Information Table 2, we have added a column to explain the processes/contributions included in each stage.

2) What is the considered silicon type (single-crystalline or multi-crystalline)? They have different efficiencies, energy demand and carbon emissions. Specifically, the Cz process in the single-silicon cell is more energy demanding and more impactful in terms of carbon emissions (<https://doi.org/10.3390/en9080622>).

Response to reviewer comments:

We consider the mix of the two dominant silicon types (single-crystalline or multi-crystalline) to calculate the emission coefficients for the PV products.

3) The USA wafer carbon emissions listed are 302.83 kgCO₂/kg wafer, while wafers produced in China cause 150.29 kg CO₂/kg wafer. Why the USA emission factors is approximately twice as that for China? On the contrary, the carbon emissions of US silicon and US solar cells are lower compared to the ones in China (79.34 kg CO₂/kg silicon compared to 129.86 kg CO₂/kg silicon and 0.79 kgCO₂eq/Wcell compared to 0.96 kgCO₂eq/ Wcell)? What is the reason for such differences?

Response to reviewer comments:

In the last version, we employed carbon emission coefficients from different data sources/literature which caused such differences. In the revision, we applied the uniform data source, i.e., Ecoinvent to avoid such differences.

5) Solar cell and PV module carbon emissions are expressed per W cell. What are the assumed efficiencies? (Please see the Fraunhofer report in which the latest commercial c-Si efficiencies are listed on pag. 29:

<https://www.ise.fraunhofer.de/content/dam/ise/de/documents/publications/studies/Photovoltaics-Report.pdf>)

Response to reviewer comments:

For the 2017 status quo, the corresponding conversion efficiencies were assumed to be 16-25% for mono-crystalline silicon technology and 14-18% for multi-crystalline silicon technology, respectively (IEA-PVPS. Trends in Photovoltaic applications 2018 (2018)).

6) The function units in Table 2 are different (kg and W). How did you combine those? What is the material usage (kg per silicon used per wafer and kg of wafer used per W cell)?

Response to reviewer comments:

The emission coefficients based on different function units in Supplement Information Table 3 (revised coefficients) were all converted to emission coefficients based on status quo price and trade value. In Methods section, it is mentioned “since the PV product trade volume in the constructed TFMs are in US\$ but not in physical quantities, the emissions coefficients measured by PV product quantity were converted into emissions per unit value in US\$ based on the prices of PV products.” The Life cycle inventory of the present study were drawn from Ecoinvent, that, 1 m² of

mono-crystalline wafer and multi-crystalline wafer use 1.07 kg and 1.14 kg silicon respectively. For multi-crystalline wafer production, the 1 m² of wafer surface is sawn into square wafers with a size 156x156 mm² (0.0243 m²) and a thickness 240 µm, and the weight is 559 g/m². For mono-crystalline wafer production, the 1 m² of wafer surface is sawn into square wafers with a size 156x156 mm² (0.0243 m²) and a thickness of 270 µm, and the weight is 629 g/m².

7) What is the thickness of the wafers?

Response to reviewer comments:

The thickness of the mono-crystalline wafer is 270 µm and the thickness of the multi-crystalline wafer is 240 µm.

8) The BOS contribution is not listed. BOS emissions factor ranges from 251-280 kgCO₂eq/kWp (considering power conversion efficiencies of 20%-18%) (see figure 4 in the following reference: <https://www.mdpi.com/1996-1073/9/8/622>)

Response to reviewer comments:

Emission coefficients of BOS and battery storage were drawn from the recommended literature and listed in Supplementary Table 4.

Reviewer comments:

Page 21 lines 499-507

Air pollutants (PM₁₀, NO_x and SO₂) of PV trade were assumed from the literature.

For China ref. n. 12 is assumed as data source, while for all the other countries Ecoinvent database is used. To be consistent it would be better to assume the same source of data.

Response to reviewer comments:

Many thanks for the kind comments and suggestion. In the revision, we focused only on carbon emission mitigation, and deleted the discussion on the air pollutant issue.

Reviewer comments:

Air pollutants are listed in Table 3 and 4 in the supplementary information.

Please clarify which countries are included in “other”.

Also, other questions are?

- Why do Europe and America have the same local pollutant emissions? The respective electricity grid mixes are different.

- What are the efficiencies assumed per W?

- Wafer thickness?

- In Table 4 (emissions of PV in China) two years are assumed as reference (2014 and 2017), while in Table 3 there is only one year. What is the reason?

- Please explain the reason for huge differences between 2014 and 2017 values in Table 4 (such as PM₁₀ from solar module: 0.103 g/W cell for 2014 compared to 0.024 g/W cell for 2017).

- It is also not clear the reason for differences between values in table 3 compared to those in table 4. As an example, NO_x associated to solar module in Europe are 0.721 kg/W module (which is similar to the value for “other” countries), while in China in 2014 are 1.389 and in 2017 are 0.215.

Response to reviewer comments:

Many thanks for the kind comments and suggestion. In the revision, we focused only on carbon emission mitigation, and deleted the discussion on the air pollutant issue.

Reviewer comments:

Page 21, lines 494-495

It is stated that “the emissions coefficients were converted into emissions per US\$”. Please provide further details on the conversion factors that are used with references and explanation.

Response to reviewer comments:

Many thanks for the kind comments and suggestion. In the revision, we have added more description about how to get the emission per US\$ in the revision, which can be found in both Methods section “**Accounting embodied carbon emissions in PV products and their global trade**” and Supplementary Information 3.1 “**Life cycle carbon emission coefficients for PV products**”.

Reviewer comments:

Page 21 lines 525-526

Please provide further details on the solar power generation potential predicted for the installed capacity increment.

Response to reviewer comments:

Many thanks for the kind comments and suggestion. In the revision, we deployed an integrated dynamic model (IMS) to simulate the power supply mix and the PV power generation potential with the installed capacity increment at country/economy level. The detailed description is shown in Methods section “**A technology-based integrated dynamic model to project power market share**” and Supplement Information 5.

Reviewer comments:

Page 22 line 532

In your second scenario, did you replace all the thermal power generation by PV?

Response to reviewer comments:

Many thanks for the kind questions. This study has assumed two scenarios for the emission reduction potential estimation. The second scenario assumed that, PV power will all be used to substitute for thermal power generation (SST), or the emission reduction parameter is the difference between the emission of thermal power generation and the PV power generation.

Reviewer comments:

Figure 2 a page 8:

the pie chart shows the CO₂ percentage for silicon, wafer, and C&M? Please provide explanation on C&M contribution? Does it stand for cell and module?

Figures 2c and 2d: the legend is difficult to read.

Response to reviewer comments:

Many thanks for the kind comments and suggestion. In the revision, we have replaced the pie charts with chord graphs, and provided detailed explanation for each figure, which is shown in Fig. 1.

Reviewer comments:

Figure 3:

Please provide explanation for local pollutant reduction from 2014 to 2017.

Response to reviewer comments:

Many thanks for the kind comments and suggestion. In the revision, we focused only on carbon emission mitigation potential of PV trade, and deleted the discussion on local pollutants reduction.

Reviewer comments:

Page 10, Lines 226-229:

Please detail the assumptions on thermal power generation replacement.

Response to reviewer comments:

Many thanks for the kind comments and suggestion. In the revision, we have added more detailed description about assumptions on thermal power generation replacement, which is in Methods section “**Accounting emissions reduction potential generated from traded solar cells and modules**”.

Reviewer comments:

Page 11 lines 265-266

It is stated that “solar power replacement of fossil fuel-fired generation also leads to local air pollutant emissions abatement”.

As stated above in this report, the contribution of balance of systems should be added because it is a crucial part of PV systems for producing electricity.

Also, electricity grids with a large penetration of PV (such as in your future scenarios) need storage systems, and their life-cycle emissions should be considered.

Please see this 2020 manuscript: <https://doi.org/10.1002/ente.201901146> and the other PV LCA literature studies on storage listed above.

Response to reviewer comments:

Many thanks for the kind comments and suggestion. In the revision, we have added BOS and storage system emissions, referring to the literature recommended to obtain BOS and storage system emission factors and calculated carbon emission. The detailed information is shown in Supplement Information 3.2 “**carbon embodied in BOS and storage system**”. In the revision, we focused only on carbon emission mitigation potential of PV trade, and deleted the discussion on local pollutants reduction.

Reviewer comments:

In the manuscript, it is used “lifecycle” as a one word, but it is more common “life cycle” as two words in the LCA literature.

Response to reviewer comments:

Many thanks for the kind comments and suggestion. We have checked the whole paper and revised “lifecycle” as “life cycle”.

Reviewer comments:

Often, it is implicitly assumed in the manuscript that silicon is the only available technologies for PV panels, and this is incorrect. Some examples are:

Lines 35-38

“Although PV power generation has nearly ‘zero emissions’ in its operation, pollutant emissions cannot be ignored when considering the whole lifecycle of PV products--from ore mining, silicon and silicon wafer processing, and solar cell and PV panel producing”

This statement implicitly assume that PV systems can be only be fabricated with silicon and it is incorrect. It should be specified that your statement refers to PV crystalline-based technologies. Although the main commercial PV technology is currently based on silicon (~94% of the market) – there are also other PV technologies commercially available, such as thin film cadmium telluride (CdTe) and copper indium gallium (di)selenide/(di) sulfide (CIGS) technologies [1]. Thin film technologies are more recent PV technologies compared to silicon-based architectures, and they offer potential benefits compared to wafer technology, including ~100x thinner absorber layers and opportunities for low cost. Thin film technologies also show reductions in life cycle carbon emissions and other environmental impacts [2]. There are also PV emerging technologies – such as perovskite PV (single junction and tandem architectures) – that have been attracting intensive attention due to notable laboratory efficiency improvements, and that may be implemented to industrial scale [3]. Their life-cycle impacts have been also evaluated in recent studies [4, 5, 6, 7, 8].

[1] Solar Power Europe. Global market outlook for solar power 2015-2019; solar power Europe. Brussels, Belgium: European Photovoltaic Industry Association; 2015. p. 2015.

[2] Fthenakis, V., Athias, C., Blumenthal, A., Kulur, A., Magliozzo, J. and Ng, D., 2020. Sustainability evaluation of CdTe PV: An update. *Renewable and Sustainable Energy Reviews*, 123, p.109776.

[3] Li, Z., Klein, T.R., Kim, D.H., Yang, M., Berry, J.J., van Hest, M.F. and Zhu, K., 2018. Scalable fabrication of perovskite solar cells. *Nature Reviews Materials*, 3(4), pp.1-20.

[4] Gong J, Darling SB, You F. Perovskite photovoltaics: Life-cycle assessment of energy and environmental impacts. *Energy Environ Sci* 2015. doi:10.1039/c5ee00615e.

[5] Celik I, Phillips AB, Song Z, Yan Y, Ellingson RJ, Heben MJ, et al. Environmental analysis of perovskites and other relevant solar cell technologies in a tandem configuration. *Energy Environ Sci* 2017;10:1874–84. doi:10.1039/c7ee01650f.

[6] Billen, P., Leccisi, E., Dastidar, S., Li, S., Lobaton, L., Spatari, S., Fafarman, A.T., Fthenakis, V.M. and Baxter, J.B., 2019. Comparative evaluation of lead emissions and toxicity potential in the life cycle of lead halide perovskite photovoltaics. *Energy*, 166, pp.1089-1096.

[7] Leccisi, E. and Fthenakis, V., 2019. Critical review of perovskite photovoltaic life cycle environmental impact studies. In 2019 IEEE 46th Photovoltaic Specialists Conference (PVSC) (Vol. 2, pp. 1-6).

[8] Celik, I., Song, Z., Cimaroli, A.J., Yan, Y., Heben, M.J. and Apul, D., 2016. Life Cycle Assessment (LCA) of perovskite PV cells projected from lab to fab. *Solar Energy Materials and Solar Cells*, 156, pp.157-169.

Lines 428-429

Please specify that it is referred only to crystalline-based technologies.

Response to reviewer comments:

Many thanks for the kind comments and suggestion. In the revision, according to reviewer suggestion. In the revision, we specified as indicated that “The present study considered and referred to the status quo PV technology composition/mix of mono-crystalline silicon (35%), multi-crystalline silicon (61%) and thin-film (4%) solar cells and modules in the PV market, or referred mainly to crystalline-based PV technologies, to perform the analysis.”

Reviewer comments:

Lines 468-470 and lines 136-137

This is a repetition. The following sentence “this study converted all the currencies to US dollars first and then converted them to 2010 US dollars for comparability” is repeated two times.

Response to reviewer comments:

Many thanks for the kind comments and suggestion. We have revised this problem and deleted the repetition sentences from the manuscript.

Peer review file, reviewer comments further–

Reviewer #1 (Remarks to the Author):

The authors have revised the paper according to the comments. I have no more comments.

Reviewer #2 (Remarks to the Author):

Summary: I think most of my initial comments hold despite the rebuttal from the authors. The core of the paper is still a difficult read, there are still too many significant digits in the numbers that distracts from the message and relative orders of magnitude, and the key messages get lost in the on slot of results. Even the abstract is barely readable. I recognize the huge amount of work that has gone into this effort including the use of two complex bottom-up energy models, the LCA analysis etc. Despite all of this arsenal, I still think a back of the envelope calculation would have given plausible results and certainly within the range of uncertainty one has about future policy considerations, the evolution of trading patterns and technology, and the future energy mix. The additional materials do help to some extent, but also raise additional questions.

- It is not clear how the SSG and SST scenarios are modeled. My interpretation is that in SSG the PV bundle substitutes for all possible non-PV electric bundles (including other renewables, nuclear, hydro etc.) and that in SST, PV only substitutes for fossil-fuel electric bundles. How is this done? and what is the justification?
- The discussion on the substitution elasticities left me perplexed. The authors state that they assume perfect substitution between imports and domestic production, but the substitution elasticity appears to be 1 (Supplementary Table 8), which indicates a rather low level of substitution. This is rendered more confusing in the sensitivity section, where the substitution elasticity range is $\pm 10\%$. If the substitution elasticity is infinity, obviously you cannot have this range. Let's infer that the authors' intent is that there is imperfect substitution between imports and domestic goods (also called the Armington assumption). In this case, a value of 1 seems low—and though a source is cited for this value, it is uniform across all importers suggesting that it was imposed as opposed to being econometrically estimated. There have been recent efforts to estimate the substitution elasticity as the HS6 level that the authors might wish to have a look at. Soderbery 2018 has a mean substitution elasticity of over 3. Fontagné et al. have a mean substitution elasticity of around 5. I would conclude that true sensitivity analysis should be much greater than the $\pm 10\%$ range and that the study's central value of 1 is at the extreme lower end of a plausible value.
- As mentioned above, the writing still needs working. My preference as well would be to write in the present tense. For example, from Page 2, "The present study aimed to explore the impacts of trade liberalization ..." could/should be written as "This study aims to explore the impacts of trade liberalization ..."

Anson Soderbery, Trade elasticities, heterogeneity, and optimal tariffs, *Journal of International Economics*, Volume 114, 2018, Pages 44-62, <https://doi.org/10.1016/j.jinteco.2018.04.008>.
Lionel Fontagné, Houssein Guimbard & Gianluca Orefice, 2019. "Product-Level Trade Elasticities," CEPII Working Paper 2019- 17, December 2019, CEPII.

Reviewer #3 (Remarks to the Author):

Thank you for the modifications to the text. My requests have been addressed.

However, have one comment regarding my question n. 7

"What is the thickness of the wafers?".

Your reply is:

"The thickness of the mono-crystalline wafer is 270 μm and the thickness of the multi-crystalline wafer is 240 μm ".

This does not reflect the state of the art. The current wafer thickness ranges from 170-180 μm .

Please see the following source, and correct it in the text where this value has been used.

R. Frischknecht, P. Stolz, L. Krebs, M. de Wild-Scholten, P. Sinha, V. Fthenakis, H. C. Kim, M. Raugei, M. Stucki, 2020, Life Cycle Inventories and Life Cycle Assessment of Photovoltaic Systems, International Energy Agency (IEA) PVPS Task 12, Report T12-19:2020.

**Title: Breaking down barriers on PV trade will facilitate global
carbon mitigation**

Formerly titled: Barriers on PV trade will impede global carbon mitigation”

Response to the reviewers’ comments

Reviewer #1:

Reviewer comments:

The authors have revised the paper according to the comments. I have no more comments.

Response to reviewer comments:

Many thanks for the comments.

Reviewer #2:

Reviewer comments:

Summary: I think most of my initial comments hold despite the rebuttal from the authors. The core of the paper is still a difficult read, there are still too many significant digits in the numbers that distracts from the message and relative orders of magnitude, and the key messages get lost in the on slot of results. Even the abstract is barely readable. I recognize the huge amount of work that has gone into this effort including the use of two complex bottom-up energy models, the LCA analysis etc. Despite all of this arsenal, I still think a back of the envelope calculation would have given plausible results and certainly within the range of uncertainty one has about future policy considerations, the evolution of trading patterns and technology, and the future energy mix.

The additional materials do help to some extent, but also raise additional questions.

Response to reviewer comments:

Many thanks for the comments and suggestion. In the revision, we have revised the manuscript by polishing the language, writing in present tense, removing redundant expression and analysis in the result section “Barriers to the PV product trade would impede global emissions reduction potential”. We sincerely hope that our revision could improve the readability of the manuscript. We have also changed the original title “Barriers on PV trade will impede global carbon mitigation” to “Breaking down barriers on PV trade will facilitate global carbon mitigation” to make the research more meaningful. The raised additional questions from the Supplementary Information

are addressed in the next responses.

Reviewer comments:

It is not clear how the SSG and SST scenarios are modeled. My interpretation is that in SSG the PV bundle substitutes for all possible non-PV electric bundles (including other renewables, nuclear, hydro etc.) and that in SST, PV only substitutes for fossil-fuel electric bundles. How is this done? and what is the justification?

Response to reviewer comments:

Many thanks for the comments. This research assumes two scenarios, including SSG and SST, for the upper and lower limits estimation of the emission reduction potential. As described in Methods section named “Accounting emissions reduction potential generated from traded solar cells and modules”, SSG represents the situation of substituting PV power for non-PV electricity in the local grid, where the carbon emission reduction potential is calculated with the average emissions coefficients of the local power grid (a mix of various energy sources, excluding PV power); and SST represents the situation that substituting PV power for fossil fuel combustion thermal power generation, and the emissions reduction potential is estimated with the average emissions coefficients of local fossil fuel combustion or thermal power plants in involved countries/economies. The contribution to carbon mitigation of traded solar cells and modules could be quantified by calculation and analysis under SSG and SST, which can provide upper and lower limits of carbon emission reduction from traded solar cells and modules application.

Reviewer comments:

The discussion on the substitution elasticities left me perplexed. The authors state that they assume perfect substitution between imports and domestic production, but the substitution elasticity appears to be 1 (Supplementary Table 8), which indicates a rather low level of substitution. This is rendered more confusing in the sensitivity section, where the substitution elasticity range is $\pm 10\%$. If the substitution elasticity is infinity, obviously you cannot have this range. Let’s infer that the authors’ intent is that there is imperfect substitution between imports and domestic goods (also called the Armington assumption). In this case, a value of 1 seems low—and though a source is cited for this value, it is uniform across all importers suggesting that it was imposed as opposed to being econometrically estimated. There have been recent efforts to estimate the substitution elasticity as the HS6 level that the authors might wish to have a look at. Soderbery 2018 has a mean substitution elasticity of over 3. Fontagné et al. have a mean substitution elasticity of around 5. I would conclude that true sensitivity analysis should be much greater than the $\pm 10\%$ range and that the study’s central value of 1 is at the extreme lower end of a plausible value.

Response to reviewer comments:

Many thanks for the comments and suggestions. We indeed made an inappropriate assumption and used inappropriate substitute elasticity value in the previous version. In this revision, we have referred to the researches you mentioned and applied the substitution elasticity of solar cell and module trade drawn from the state of the art research, which are econometrically estimated. We have then explained this point in the Methods section titled “Trade barrier scenarios and computable partial equilibrium model simulation” that, “Supply, demand and substitute elasticities of solar cell and module trade are obtained from current research, which are econometrically

estimated (56) (see Supplementary Table 8)", and in the Supplementary 6.1 that, "there have been recent efforts to estimate the substitution elasticity as the HS6 level, e.g., Soderbery (38) has a mean substitution elasticity of over 3, Fontagné et al. (39) have a substitution elasticity centered around 5. Elasticities of solar cells and modules trade for the current study are referred from the latest study, which adopts a double logarithmic model to estimate Armington substitution elasticity of solar cell and module trade tailored for GSIM model (40). (see Supplementary Table 8)." Then we re-ran GSIM model and updated all the simulation results in the section titled "Barriers to the PV product trade would impede global emissions reduction potential", including redrawing Figure 7. The relevant elaboration of the simulation results and sensitivity analysis in the Supplementary Information 6.2, 6.3 are also revised accordingly.

Reviewer comments:

As mentioned above, the writing still needs working. My preference as well would be to write in the present tense. For example, from Page 2, "The present study aimed to explore the impacts of trade liberalization ..." could/should be written as "This study aims to explore the impacts of trade liberalization ..."

Response to reviewer comments:

Many thanks for the comments and suggestions. We have carefully revised the manuscript to write in the present tense. For example, in Page 2, "The present study aims to explore the impacts of trade liberalization...", in Page 3, "This study focuses on the carbon emissions reduction potential...", and in Page 4, "East Asia ranks as the top PV product exporter...". Please see other revisions in the revised manuscript.

Reviewer #3:

Reviewer comments:

Thank you for the modifications to the text. My requests have been addressed.

However, have one comment regarding my question n. 7

"What is the thickness of the wafers?".

Your reply is:

"The thickness of the mono-crystalline wafer is 270 μm and the thickness of the multi-crystalline wafer is 240 μm ".

This does not reflect the state of the art. The current wafer thickness ranges from 170-180 μm . Please see the following source, and correct it in the text where this value has been used.

R. Frischknecht, P. Stolz, L. Krebs, M. de Wild-Scholten, P. Sinha, V. Fthenakis, H. C. Kim, M. Raugei, M. Stucki, 2020, Life Cycle Inventories and Life Cycle Assessment of Photovoltaic Systems, International Energy Agency (IEA) PVPS Task 12, Report T12-19:2020.

Response to reviewer comments:

Many thanks for the comments.

We have revised the corresponding text in the Supplementary Information 3.1 which is read as "Thus, based on LCA results from Ecoinvent database, this study considers improved technologies, including reduced silicon wafer thickness (the thickness of the mono-crystalline wafer is 270 μm

and that of the multi-crystalline wafer is 240 μm according to Ecoinvent database, and the state of the art literature shows that the current wafer thickness ranges from 170-180 μm (3).

Peer review file, reviewer comments further--

Reviewer #2 (Remarks to the Author):

The authors have addressed many of my concerns, though it is still a tough read with respect to the grammar and most of the numbers in the text and tables contain too many significant digits that are a distraction--round numbers are highly recommended, particularly regarding future indicators where uncertainty is particularly high. For example, a key sentence in the abstract is "From 2017 to 2060, solar power generation will result in an approximately 51.47-182.68 GtCO_{2e} net reduction in BAU.", which would be more readable as "Solar power generation would result in a reduction of emissions in a range of 50-180 GtCO_{2e} between 2017 and 2060.". In fact, the use of the word 'approximately' seems misplaced!

The sensitivity analysis is not very revealing--a range of plus or minus 10% is narrow and I suspect within this range the model's behavior is nearly linear. The authors could have used the standard errors from the econometric studies to look at a range of plus or minus 1 or 2 times the standard deviation.

Reviewer #3 (Remarks to the Author):

The requests and comments of this reviewer have been addressed.

**Title: Breaking down barriers on PV trade will facilitate global
carbon mitigation**

Response to the reviewers' comments

REVIEWERS' COMMENTS

Reviewer #2 (Remarks to the Author):

- The authors have addressed many of my concerns, though it is still a tough read with respect to the grammar and most of the numbers in the text and tables contain too many significant digits that are a distraction--round numbers are highly recommended, particularly **regarding future indicators** where uncertainty is particularly high. For example, a key sentence in the abstract is "From 2017 to 2060, solar power generation will result in an approximately 51.47-182.68 GtCO_{2e} net reduction in BAU." , which would be more readable as **"Solar power generation would result in a reduction of emissions in a range of 50-180 GtCO_{2e} between 2017 and 2060."** . In fact, the use of the word 'approximately' seems misplaced!

Response:

Many thanks for the kind suggestions. We thus have revised the key sentence in the abstract as "Solar power generation will result in a reduction of emissions in a range of 50-180 gigatons of carbon dioxide equivalent (GtCO_{2e}) between 2017 and 2060 in business as usual (BAU)". All the numbers with two significant digits in the abstract and introduction section have been revised to round numbers.

We would like to maintain the two significant digits in the rest sections to reflect the stringency of the calculation and simulation. Since the scale of the units for the numbers are large enough, e.g., GW, TWh, GtCO_{2e}, etc., we think keeping two significant digits would be acceptable.

- The **sensitivity** analysis is not very revealing--a range of plus or minus 10% is narrow and I suspect within this range the model's behavior is nearly linear. The authors could have used the standard errors from

the econometric studies to look at a range of plus or minus 1 or 2 times the standard deviation.

Response:

Since we did not apply a statistical or econometric analysis here, and thus the data could not support such analysis, to address this issue, we expanded the range of elasticity change to plus and minus 20%. The changing rate (SenSE) of import in most countries/economies will be within 4.0% under TBS0, will be within 2.0% under TBS2, and will be almost neglectable under TBS1. The SenSE of consumption in most countries/economies will be within 2.0% under TBS0, will be within 1.0% under TBS2, and will be almost neglectable under TBS1. The lack of sensitivity of solar cells and modules trade and consumptions to substitute elasticity provides support of robustness and reliability of the GSIM model simulation results, and the subsequent estimation of the policy shock impacts on the trade related solar PV power generation potential and carbon emission reduction of PV application.

Reviewer #3 (Remarks to the Author):

- The requests and comments of this reviewer have been addressed.

Response:

Many thanks.